# Demonstration Selection for In-Context Learning via Reinforcement Learning

Xubin Wang [1 2 3]   Jianfei Wu [3]   Yichen Yuan [1]   Deyu Cai [1]   Mingzhe Li [1]   Weijia Jia [1 3]

## Abstract

Diversity in demonstration selection is critical for enhancing model generalization by enabling broader coverage of structures and concepts. Constructing appropriate demonstration sets remains a key research challenge. This paper introduces the Relevance-Diversity Enhanced Selection (RDES), an innovative approach that leverages reinforcement learning (RL) frameworks to optimize the selection of diverse reference demonstrations for tasks amenable to in-context learning (ICL), particularly text classification and reasoning, in few-shot prompting scenarios. RDES employs frameworks like Q-learning and a PPO-based variant to dynamically identify demonstrations that maximize both diversity (quantified by label distribution) and relevance to the task objective. This strategy ensures a balanced representation of reference data, leading to improved accuracy and generalization. Through extensive experiments on multiple benchmark datasets, including diverse reasoning tasks, and involving 14 closed-source and open-source LLMs, we demonstrate that RDES significantly enhances performance compared to ten established baselines. Our evaluation includes analysis of performance across varying numbers of demonstrations on selected datasets. Furthermore, we investigate incorporating Chain-of-Thought (CoT) reasoning, which further boosts predictive performance. The results highlight the potential of RL for adaptive demonstration selection and addressing challenges in ICL.

[1]BNU-BNBU Institute of Artificial Intelligence and Future Networks, Beijing Normal-Hong Kong Baptist University, Zhuhai, China [2]Hong Kong Baptist University, Hong Kong, China [3]Beijing Normal University at Zhuhai, Zhuhai, China. Correspondence to: Weijia Jia <jiawj@bnu.edu.cn>, Xubin Wang <wangxubin@ieee.org>.

*Proceedings of the $42^{nd}$ International Conference on Machine Learning*, Vancouver, Canada. PMLR 267, 2025. Copyright 2025 by the author(s).

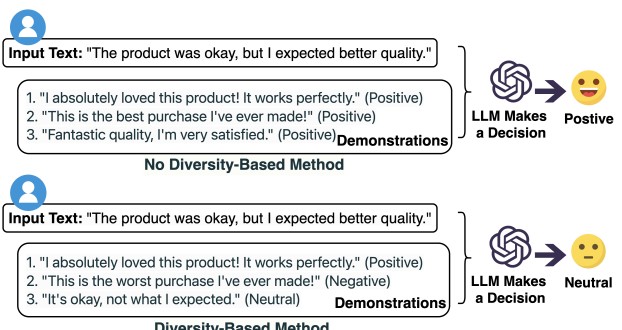

*Figure 1.* An example shows how a diversity-based demonstration method works. In this example, the diversity-based method helps the model recognize that the input text expresses a sentiment that is neither strongly positive nor negative, while the no diversity-based method may lead to an inaccurate positive classification due to its lack of varied demonstrations.

## 1. Introduction

LLMs have demonstrated exceptional capabilities across a wide array of NLP tasks, including text annotation (Wu et al., 2025), question answering (Shao et al., 2023), and dialogue generation (Hu et al., 2023). These models leverage extensive corpora of textual data to learn rich representations, which empower them to perform reasoning with high accuracy (Devlin et al., 2018; Radford et al., 2019; Brown et al., 2020). However, as the size and complexity of these models continue to expand, enhancing their reasoning capabilities becomes increasingly crucial. Effective reasoning is essential for tasks that demand logical reasoning, commonsense understanding, and contextual awareness (Marcus, 2020; Bender et al., 2021). The ability to reason effectively not only improves the performance of LLMs in existing applications but also expands their potential for novel use cases that demand a deeper understanding of language and context.

In the realm of few-shot learning, in-context learning (ICL) has emerged as a promising approach to enhance reasoning in LLMs (Dong et al., 2024). ICL utilizes LLMs, such as those based on the GPT architecture, to perform reasoning by providing a carefully curated set of demonstrations as context, rather than relying solely on extensive model retraining (Brown et al., 2020; Gao et al., 2021; Zhang et al.,

2022). This methodology allows LLMs to leverage their inherent capabilities for understanding and processing text, making them particularly suitable for tasks with limited labeled data. However, the effectiveness of ICL is contingent upon the selection of appropriate and representative demonstrations from the knowledge base to serve as contextual references during reasoning on test data. This critical aspect of few-shot learning is often overlooked in existing literature (Wang et al., 2020; Song et al., 2023). The careful selection of demonstrations is essential, as it directly influences the model's ability to generalize and perform accurately in novel situations.

Despite the promise of ICL, a significant challenge persists in selecting the most relevant and diverse demonstrations from the knowledge base to optimize reasoning performance. Traditional methods of demonstration selection often prioritize similarity, which can inadvertently overlook the importance of diversity in capturing the full spectrum of the data distribution (Song et al., 2023). This oversight may lead to biased representations that do not generalize well to unseen data, ultimately hindering the predictive accuracy of LLMs (Wang et al., 2020; Liu & Lapata, 2019). Furthermore, conventional selection techniques typically employ fixed strategies that fail to dynamically adapt to the specific requirements of the reasoning task at hand (Wang et al., 2020; Song et al., 2023). This rigidity can limit the effectiveness of ICL, as the selected demonstrations may not align optimally with the context or nuances of the task, further exacerbating the challenges in achieving robust reasoning performance.

The core motivation behind Relevance-Diversity Enhanced Selection (RDES) is to enhance performance in tasks amenable to ICL, particularly text classification and reasoning. This is achieved by selecting demonstrations that maximize relevance while ensuring diversity. While similarity-based selection improves accuracy, it risks overfitting, whereas diversity promotes generalization, particularly in few-shot learning. By framing demonstration selection as a sequential decision-making problem, RDES effectively leverages reinforcement learning (RL) frameworks, including Q-learning and a Proximal Policy Optimization (PPO)-based variant, to balance exploration and exploitation, improving model robustness and adaptability. RDES is evaluated and compared against two baseline categories: prompt engineering and demonstration selection. Prompt engineering focuses on crafting effective prompts, while demonstration selection aims to identify the most informative references. Our main contributions are as follows:

- We introduce RDES, a RL-based framework that dynamically selects demonstrations to enhance performance and model robustness for tasks amenable to ICL.

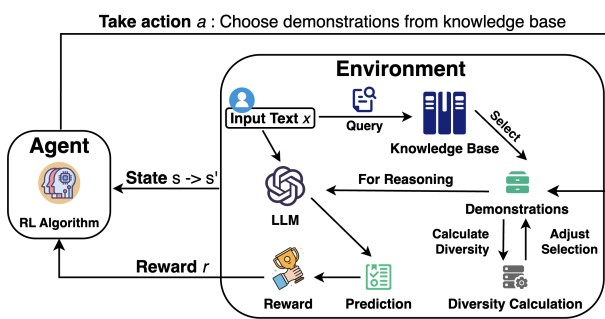

*Figure 2.* The RDES framework is an adaptive RL approach for few-shot ICL demonstration selection in LLMs. It employs a RL-based agent to dynamically balance the relevance and diversity of selected examples, guided by a reward function that incorporates a label distribution diversity score. This strategy enhances classification accuracy and generalization by mitigating overfitting associated with pure similarity-based methods. The framework involves an Agent interacting with an Environment (including a Knowledge Base and the LLM) to learn an optimal selection policy.

- Our RL approach (including Q-learning and a PPO variant) optimizes selection by balancing relevance and diversity, mitigating overfitting and improving generalization.

- RDES integrates seamlessly with advanced reasoning techniques, such as Chain-of-Thought (CoT) prompting, further enhancing LLM reasoning capabilities.

- We perform comprehensive evaluations against ten baseline methods across multiple benchmark datasets, showcasing significant improvements in diverse tasks with 14 closed-source and open-source LLMs.

In summary, RDES advances few-shot learning in NLP by addressing ICL's key challenges through adaptive demonstration selection. By jointly optimizing relevance and diversity, it enhances LLMs' performance on tasks amenable to ICL, particularly classification and reasoning. The schematic framework of RDES is illustrated in Figure 2.

## 2. Related Work

### 2.1. In-Context Learning

ICL has emerged as a transformative paradigm in NLP, particularly with the advent of LLMs, enabling models to adapt to new tasks by conditioning on a small set of demonstrations within the input context, thus eliminating the need for extensive retraining. The seminal work by Brown *et al.* introduced ICL with the GPT-3 model, demonstrating that LLMs can effectively perform a wide range of tasks by being exposed to a few exemplars in the prompt (Brown

et al., 2020). Subsequent research has further explored ICL's capabilities, highlighting its effectiveness in few-shot and zero-shot learning scenarios (Wei et al., 2021; Liu et al., 2023). However, the effectiveness of ICL is critically contingent upon the selection of appropriate and representative demonstrations from the knowledge base to serve as contextual references, an aspect often overlooked in existing literature. The careful selection of demonstrations is essential, as it directly influences the model's ability to generalize and perform accurately in novel situations. Recent studies emphasize the need for effective demonstration selection strategies to enhance ICL outcomes, especially in contexts with limited labeled data (Zhang et al., 2022; Min et al., 2022; Ye et al., 2023; Bai et al., 2024). Approaches such as Determinantal Point Process (DPP) methods and Iterative Demonstration Selection (IDS) have highlighted the importance of selecting a diverse and relevant subset of demonstrations to optimize model performance (Wang et al., 2024; Qin et al., 2024).

### 2.2. Demonstration Selection Techniques

Effective demonstration selection is crucial for ICL success. Traditional methods often relied on heuristics or statistical measures to identify representative examples, including selecting informative examples to minimize uncertainty, a concept central to active learning (Settles, 2009). Recently, the importance of diversity in the selection process has gained prominence, with research showing that a diverse set of examples can improve generalization capabilities (Sener & Savarese, 2018). Techniques such as clustering, coverage-based selection (e.g., BERTScore-Recall), and representative sampling are employed to ensure chosen examples cover a broad range of the input space (Zhang et al., 2023; Gupta et al., 2023). These approaches aim to enhance model performance, particularly in compositional tasks where diverse demonstrations offer better coverage. Skill-based few-shot selection methods, like Skill-KNN, optimize example selection by eliminating irrelevant features (An et al., 2023). Methods such as (Yang et al., 2023) prioritize diversity statically, while (Mavromatis et al., 2023) selects based on uncertainty and diversity without training a policy. The work (Levy et al., 2023) also aligns with the core idea of using diversity to enhance generalization. Calibration techniques (Zhao et al., 2021) focus on correcting biases in demonstration selection, which overlaps with the effort to ensure diverse label coverage.

### 2.3. RL in Demonstration Selection

RL has emerged as a promising framework for optimizing demonstration selection across diverse machine learning tasks. Unlike static approaches, RL allows selection policies to adapt iteratively based on feedback from the model's performance. Prior work like RetICL leveraged RL to op-

timize the selection and sequencing of examples for ICL (Scarlatos & Lan, 2023). Similarly, (Zhang et al., 2022) formulated the demonstration selection problem as a sequential decision-making task, utilizing a Q-learning framework to enhance example selection. The proposed RDES method builds upon these foundational studies, sitting at the intersection of ICL, demonstration selection, and RL-based post-training for LLMs. However, while prior RL methods like (Scarlatos & Lan, 2023) and (Zhang et al., 2022) primarily focused on objectives such as relevance or uncertainty, RDES uniquely focuses on the dual objectives of diversity and relevance, explicitly aiming to optimize both by incorporating a diversity score into the reward function. RDES employs a Q-learning framework and also explored a PPO-based variant (RDES/PPO) which showed competitive results. This combination of a RL framework with an explicit label distribution-based diversity metric to achieve dynamic demonstration selection that balances relevance and diversity is highlighted as novel. RDES integrates seamlessly with advanced reasoning techniques, such as CoT prompting, with the RDES/C variant incorporating CoT reasoning to further enhance predictive performance. While RL from Human Feedback (RLHF) fine-tunes model weights for alignment, RDES optimizes input selection while keeping the model fixed, presenting a lighter-weight alternative for post-training improvements. By learning an adaptive policy per query, RDES can better optimize selection compared to methods like (Yang et al., 2023) and (Mavromatis et al., 2023).

## 3. Methodology

The RDES framework tackles the demonstration selection challenge using a principled RL approach that jointly optimizes for relevance and diversity. This section systematically presents our methodology in four progressive components: (1) formal problem formulation as a Markov Decision Process, (2) dual optimization strategies using Q-learning and PPO, and (3) implementation details.

### 3.1. Theoretical Foundations

#### 3.1.1. RL FORMULATION

RL provides a natural framework for sequential decision making in demonstration selection. We model the interaction between the selection policy and language model as an iterative process where the policy learns to construct optimal demonstration sets through trial-and-error interactions.

#### 3.1.2. MARKOV DECISION PROCESS CONSTRUCTION

The demonstration selection process is formalized as a finite-horizon MDP $\mathcal{M} = (\mathcal{S}, \mathcal{A}, \mathcal{P}, \mathcal{R}, \gamma)$ with the following components:

- **State Space** ($\mathcal{S}$): Captures the complete decision context through four components:

  - Textual features: $\phi_x(x_t) \in \mathbb{R}^{d_x}$ (TF-IDF vector of input text)
  - Demonstration memory: $\phi_E(E_t) \in \mathbb{R}^{d_e}$ (Aggregated embeddings of selected examples)
  - Prediction history: $\phi_y(\hat{y}_t) \in \mathbb{R}^{|\mathcal{Y}|}$ (One-hot encoded previous predictions)
  - Diversity tracking: $D_t = \frac{|\mathcal{L}(E_t)|}{k} \in [1]$ (Normalized label diversity)

The state embedding is constructed by concatenating these four distinct components:

$$\phi(s_t) = \phi_x(x_t) \oplus \phi_E(E_t) \oplus \phi_y(\hat{y}_t) \oplus \phi_D(D_t) \quad (1)$$

where $\oplus$ denotes vector concatenation, and each $\phi.$ represents an embedding for the respective component.

- **Action Space** ($\mathcal{A}$): Discrete selection over candidate demonstrations $\mathcal{K}$, with action $a_t \in \{1, ..., |\mathcal{K}|\}$ indicating the chosen example index from the knowledge base.

- **Transition Dynamics** ($\mathcal{P}$): Deterministic state updates through demonstration set modification. When action $a_t$ (selecting candidate $k_{a_t}$) is taken in state $s_t = (x_t, E_t, \hat{y}_t, D_t)$, the next state $s_{t+1}$ becomes:

$$s_{t+1} = f(s_t, a_t) = (x_t, E_t \cup \{k_{a_t}\}, \hat{y}_{t+1}, D_{t+1}) \quad (2)$$

where $\hat{y}_{t+1}$ is the new prediction based on the updated example set and $D_{t+1}$ is the new diversity score.

- **Reward Function** ($\mathcal{R}$): A multi-objective reward balancing prediction accuracy and diversity gain:

$$\mathcal{R}(s_t, a_t) = \underbrace{\mathbb{I}(y_{\text{true}} = \hat{y}_t)}_{\text{Accuracy}} + \lambda \underbrace{(D_{t+1} - D_t)}_{\text{Diversity Improvement}} \quad (3)$$

where $\mathbb{I}(\cdot)$ is the indicator function, $y_{\text{true}}$ is the true label, $\hat{y}_t$ is the prediction at step $t$, $D_t$ is the diversity at step $t$, $D_{t+1}$ is the diversity after adding the selected example, and $\lambda$ controls the exploration-exploitation tradeoff. The diversity coefficient $\lambda$ adapts during training via an annealing schedule:

$$\lambda(t) = \lambda_{\min} + (\lambda_{\max} - \lambda_{\min})e^{-\eta t} \quad (4)$$

This schedule prioritizes early exploration of diverse examples before focusing on accuracy.

- **Discount Factor** ($\gamma$): $\gamma \in [0, 1)$ emphasizes immediate rewards, which is suitable for finite-horizon few-shot learning scenarios where a fixed number of examples are selected.

## 3.2. Optimization Framework

The RDES framework employs two primary RL algorithms to handle different complexities of the state space and computational resources.

### 3.2.1. Q-LEARNING APPROACH

Q-learning provides a model-free solution for learning demonstration selection strategies through temporal difference updates. This approach is particularly effective for environments with relatively small or discretizable state spaces. The action-value function $Q : \mathcal{S} \times \mathcal{A} \to \mathbb{R}$ estimates the expected cumulative rewards starting from state $s$, taking action $a$, and subsequently following the optimal policy:

$$Q(s, a) = \mathbb{E}\left[ \sum_{\tau=t}^{T} \gamma^{\tau - t} r_\tau \mid s_t = s, a_t = a \right] \quad (5)$$

The Q-values are updated via the standard Q-learning rule:

$$Q(s, a) \leftarrow Q(s, a) + \alpha \left[ r + \gamma \max_{a'} Q(s', a') - Q(s, a) \right] \quad (6)$$

where $\alpha$ is the learning rate, $r$ is the immediate reward, $s'$ is the next state, and the $\max_{a'} Q(s', a')$ term represents the estimated value of the next state under the greedy policy. Key implementation aspects for applying tabular Q-learning include state discretization, for example, through TF-IDF feature binning, and using an $\epsilon$-greedy exploration strategy with exponential decay ($\epsilon$-annealing) to balance exploration and exploitation. Tabular Q-value storage is used, with sparse updates focusing on visited state-action pairs.

### 3.2.2. PROXIMAL POLICY OPTIMIZATION VARIANT

For high-dimensional state spaces where tabular methods are infeasible, we implement an actor-critic architecture utilizing PPO. This approach uses neural networks to approximate the policy and value functions.

- **Policy Network** ($\pi_\theta$): A neural network with parameters $\theta$ that produces demonstration selection probabilities for each action given a state:

$$\pi_\theta(a|s) = \text{softmax}(W_2 \sigma(W_1 \phi(s) + b_1) + b_2) \quad (7)$$

where $\phi(s)$ is the state embedding, $\sigma$ is a non-linear activation function, and $W_i, b_i$ are learned weights and biases.

- **Value Network** ($V_\psi$): A neural network with parameters $\psi$ that estimates the value function of a state, representing the expected cumulative reward from that state:

$$V_\psi(s) = W_4 \sigma(W_3 \phi(s) + b_3) + b_4 \quad (8)$$

where $\phi(s)$, $\sigma$, $W_i$, $b_i$ are similarly defined.

- **Optimization Objective**: PPO optimizes a clipped surrogate objective function that encourages moderate policy updates, ensuring stability. The overall objective function for training the policy ($\theta$) and value ($\psi$) networks combines the clipped surrogate loss, a value function loss, and an entropy bonus:

$$\mathcal{L}(\theta, \psi) = \mathbb{E}_t \left[ \mathcal{L}_t^{\text{CLIP}}(\theta) - c_1 \mathcal{L}_t^{\text{VF}}(\psi) + c_2 \mathcal{L}_t^{\text{ENT}}(\theta) \right] \tag{9}$$

where $\mathbb{E}_t$ denotes the empirical average over a batch of trajectories, $c_1$ and $c_2$ are coefficients controlling the weights of the value and entropy terms, respectively. The components are defined as:

$$\mathcal{L}_t^{\text{CLIP}}(\theta) = \min \left( r_t(\theta) A_t, \text{clip}\left( r_t(\theta), 1 \pm \epsilon \right) A_t \right)$$
$$\mathcal{L}_t^{\text{VF}}(\psi) = (V_\psi(s_t) - \hat{R}_t)^2$$
$$\mathcal{L}_t^{\text{ENT}}(\theta) = -\sum_a \pi_\theta(a|s_t) \log \pi_\theta(a|s_t)$$

Here, $r_t(\theta) = \frac{\pi_\theta(a_t|s_t)}{\pi_{\theta_{\text{old}}}(a_t|s_t)}$ is the probability ratio between the new and old policies, $A_t$ is the advantage estimate (e.g., using Generalized Advantage Estimation - GAE), $\text{clip}(\cdot, 1 \pm \epsilon)$ clips the ratio to be within $[1 - \epsilon, 1 + \epsilon]$, $\hat{R}_t$ is the estimated return, and $\mathcal{L}_t^{\text{ENT}}$ is the entropy of the policy distribution at state $s_t$, which encourages exploration.

### 3.3. Algorithmic Implementation

#### 3.3.1. UNIFIED TRAINING PARADIGM

The core training procedure shared by both Q-learning and PPO approaches is formalized in Algorithm 1. The algorithm iteratively samples test inputs, selects a set of demonstrations (initially based on relevance, then potentially adjusted), uses these to prompt the LLM for a prediction, calculates the diversity of the selected set, encodes the current decision context into a state, selects an action (which corresponds to choosing an example, although the algorithm abstractly shows $(s, a, r)$ for policy update, implying the action leads to state transition and reward), calculates the reward based on prediction accuracy and diversity change, and updates the policy parameters using the chosen RL algorithm.

#### 3.3.2. STATE REPRESENTATION DETAILS

As detailed in the MDP construction, the state embedding $\phi(s_t)$ is a concatenation of four components: the TF-IDF vector of the input text, aggregated embeddings of selected examples, a representation of the prediction history, and the normalized label diversity. This comprehensive representation provides the RL agent with sufficient context to make informed selection decisions.

---

**Algorithm 1** RDES Training Framework

**Require:** Knowledge base $\mathcal{K}$, Test inputs $\mathcal{D}_{\text{test}}$, LLM $\mathcal{M}$, RL algorithm $\mathcal{A}$
1: Initialize selection policy $\pi$ (Q-table or neural networks)
2: Precompute TF-IDF vectors for each sample $x_i \in \mathcal{D}_{\text{test}}$ and for knowledge base $\mathcal{K}$
3: **for** $i = 1$ to $N$ **do**
4:   Sample input $x_i \in \mathcal{D}_{\text{test}}$
5:   Select demonstrations $E$ with initial candidates based on relevance (e.g., top-$k$ TF-IDF matches from $\mathcal{K}$) and apply diversity adjustment.
6:   Generate prompt $p = \text{Format}(x_i, E)$
7:   Obtain prediction $\hat{y} = \mathcal{M}(p)$
8:   Compute diversity score $D = \frac{|\mathcal{L}(E)|}{k}$
9:   Encode state $s = \phi(x_i, E, \hat{y}, D)$
10:   Select action $a \sim \pi(s)$ (example index from $\mathcal{K}$)
11:   Calculate reward $r = \mathbb{I}(y_{\text{true}} = \hat{y}) + \lambda(D_{\text{new}} - D_{\text{old}})$
12:   Update policy parameters $\theta$ using $\mathcal{A}$ with $(s, a, r)$
13: **end for**
14: **Return:** Optimized policy $\pi^*$

---

#### 3.3.3. PROMPTING STRATEGIES

To enhance the performance of the LLM in few-shot settings, we employ two distinct prompting strategies using the selected examples:

- **Standard Prompting:** This strategy constructs a prompt by concatenating the input text, the selected demonstrations (input-output pairs), and the set of possible labels. The LLM is then asked to predict the probability of a label $y$ given this prompt structure:

$$p(y|x, E) = \mathbb{P}_{\text{LM}} \left( y \mid \text{Prompt: } x, \{(\tilde{x}_i, \tilde{y}_i)\}_{i=1}^k, \mathcal{Y} \right) \tag{10}$$

where $x$ is the input text, $\{(\tilde{x}_i, \tilde{y}_i)\}_{i=1}^k$ are the $k$ selected demonstrations, and $\mathcal{Y}$ is the set of possible labels.

- **CoT Prompting:** This strategy incorporates CoT reasoning into the prompt, allowing the LLM to generate intermediate reasoning steps before producing the final label. This is formulated as marginalizing over possible reasoning chains $\mathcal{R}$:

$$p(y|x, E) = \sum_{r \in \mathcal{R}} \mathbb{P}_{\text{LM}}(r|x, E) \cdot \mathbb{P}_{\text{LM}}(y|x, E, r) \tag{11}$$

The model first computes the probability of a reasoning chain $r$ given the input and demonstrations, then the probability of the label $y$ conditioned on the input, demonstrations, and the generated reasoning chain.

# 4. Experiments

## 4.1. Datasets

In our framework evaluation, we utilize four widely recognized datasets that encompass a diverse range of domains and intents. The BANKING77 dataset (Casanueva et al., 2020) provides a comprehensive set of intents specifically relevant to the banking sector. Additionally, the HWU64 (Liu et al., 2021) and LIU54 (Liu et al., 2021) datasets offer extensive multi-domain coverage, making them particularly valuable for comparative analysis. We also include the CLINC150 dataset (Larson et al., 2019), which further enriches our evaluation framework. To better align our evaluation with real-world application scenarios, we employed a challenge set sampling strategy, drawing on the principles outlined in (Lu et al., 2024). This approach allowed us to select a demanding subset from the original test splits based on the precision margin, ensuring a rigorous assessment of our model's performance. To further assess the generalizability and effectiveness of RDES on tasks requiring more complex reasoning, we also conducted additional experiments on challenging benchmarks. These include subsets of BigBenchHard (Suzgun et al., 2023) (specifically, boolean expressions and web of lies), GSM-8K (Cobbe et al., 2021), and SST5 (Socher et al., 2013). These datasets require advanced reasoning capabilities from the LLMs and were used to evaluate how RDES performs in more complex scenarios compared to baselines.

## 4.2. Compared Methods

In this study, we evaluate ten baseline approaches for their effectiveness in various classification tasks, categorizing them into two main groups: Prompt Engineering Methods and Demonstration Selection Methods. The Prompt Engineering Methods manipulate prompt structures to enhance model understanding and decision-making, including Zero-Shot Prompting (ZS) (Radford et al., 2019), which tests generalization without prior demonstrations; Knowledge Prompting (KP) (Liu et al., 2022), which provides contextual information to improve accuracy; Least-to-Most Prompting (L2M) (Zhou et al., 2023), which breaks tasks into manageable steps; Chain of Thought (CoT) prompting (Wei et al., 2022), which encourages step-by-step reasoning; and Self-Refine (SF) (Madaan et al., 2024), which allows the model to iteratively critique and refine its solutions. The Demonstration Selection Methods utilize selected demonstrations to inform predictions, including Few-Shot Prompting (FS) (Lu et al., 2024), which enhances predictions through a limited number of text-label pairs; Few-Shot with CoT (FSC) (Lu et al., 2024), which combines demonstrations with explanations; Active Demonstration Selection (AES) (Zhang et al., 2022), which iteratively selects relevant demonstrations to improve learning efficiency; Representative Demonstration Selection

(RDS) (Yang et al., 2023), which identifies diverse subsets of demonstrations for better generalization; and Adaptive Demonstration Selection for In-Context Learning (ADA) (Mavromatis et al., 2023), which focuses on uncertain cases to enhance robustness and adaptability. Together, these methods leverage different aspects of prompting and demonstration selection to improve classification accuracy and response quality.

## 4.3. LLMs Used in Experiments

To evaluate our method, we employ a diverse set of LLMs, including both closed-source and open-source models. Closed-source models such as GPT-3.5-turbo, Doubao-lite-4k, Doubao-pro-4k, and Hunyuan-lite, developed by industry leaders, offer strong NLP capabilities for tasks like content generation and long-context understanding (Zhang et al., 2024; Team, 2024a; Cloud, 2024). In contrast, open-source models—including Gemma-2-2B, Gemma-2-9B, LLaMA-3.2-1B, LLaMA-3.2-3B, LLaMA-3-8B, Qwen-2.5-7B, Qwen-2.5-14B, and Qwen-1.5-72B—enable greater customization and adaptability. The Gemma series emphasizes efficiency, the Qwen series excels in scalability, and the LLaMA series is known for strong benchmark performance (Team et al., 2024; Bai et al., 2023; Yang et al., 2024; Dubey et al., 2024). The open-source nature of these models fosters innovation and wider experimentation compared to proprietary alternatives. Our primary experiments on the four classification benchmarks (BANKING77, HWU64, CLINC150, LIU54) were conducted using RDES/B (our base version) and RDES/C (RDES with CoT reasoning), both of which are based on the Q-learning framework detailed in Section 3.2.1. Furthermore, we conducted additional experiments on challenging benchmarks such as subsets of BigBenchHard (boolean expressions and web of lies), GSM-8K, and SST5. For these supplementary experiments, we specifically utilized models known for their strong performance in complex tasks: Qwen-25-72B (Team, 2024b) and DeepSeek-R1-32B (DeepSeek-AI, 2025). Crucially, in these supplementary experiments, we evaluated not only the Q-learning based variants (RDES/B and RDES/C) but also a PPO-based variant, denoted as RDES/PPO. The results and analysis using these models and RDES variants on the challenging reasoning benchmarks are presented in Section 4.5.

## 4.4. Reasoning Performance Analysis

This section presents a comprehensive evaluation of the reasoning accuracy of both closed-source and open-source LLMs using a range of prompt engineering and demonstration selection techniques across four benchmark datasets: BANKING77, CLINC150, HWU64, and LIU54. The evaluation focuses on popular closed-source models such as GPT-3.5-turbo, Doubao-lite-4k, Doubao-pro-4k, and Hunyuan-

*Table 1.* Performance comparison of methods designed to boost LLM reasoning across various datasets on closed-source LLMs, with a focus on accuracy. The RDES/B denotes the base version, while RDES/C indicates the version enhanced with CoT reasoning.

| Datasets | Models | Prompt Engineering Methods | | | | | Demonstration Selection Methods | | | | | Ours | |
|---|---|---|---|---|---|---|---|---|---|---|---|---|---|
| | | ZS | KP | L2M | CoT | SF | FS | FSC | AES | RDS | ADA | RDES/B | RDES/C |
| **BANKING77** | GPT-3.5-turbo | 0.340 | 0.240 | 0.260 | 0.200 | 0.380 | 0.520 | 0.320 | 0.260 | 0.240 | 0.360 | 0.767 | **0.858** |
| | Doubao-lite-4k | 0.300 | 0.300 | 0.300 | 0.320 | 0.300 | 0.500 | 0.360 | 0.300 | 0.280 | 0.400 | 0.750 | **0.830** |
| | Doubao-pro-4k | 0.500 | 0.400 | 0.500 | 0.480 | 0.600 | 0.540 | 0.540 | 0.700 | 0.680 | **0.900** | 0.838 | 0.888 |
| | Hunyuan-lite | 0.300 | 0.233 | 0.433 | 0.200 | 0.300 | 0.233 | 0.133 | 0.320 | 0.320 | 0.600 | 0.593 | **0.775** |
| | **Average** | 0.360 | 0.293 | 0.373 | 0.300 | 0.395 | 0.448 | 0.338 | 0.395 | 0.380 | 0.565 | 0.737 | **0.838** |
| **CLINC150** | GPT-3.5-turbo | 0.460 | 0.420 | 0.400 | 0.480 | 0.460 | 0.600 | 0.380 | 0.300 | 0.380 | 0.720 | 0.845 | **0.949** |
| | Doubao-lite-4k | 0.700 | 0.600 | 0.600 | 0.700 | 0.500 | 0.680 | 0.440 | 0.680 | 0.660 | 0.700 | 0.825 | **0.927** |
| | Doubao-pro-4k | 0.660 | 0.680 | 0.620 | 0.700 | 0.700 | 0.800 | 0.640 | 0.680 | 0.640 | 0.900 | 0.938 | **0.961** |
| | Hunyuan-lite | 0.633 | **0.800** | 0.767 | 0.700 | 0.633 | 0.467 | 0.500 | 0.480 | 0.620 | **0.800** | 0.730 | 0.772 |
| | **Average** | 0.613 | 0.625 | 0.597 | 0.645 | 0.573 | 0.637 | 0.490 | 0.535 | 0.575 | 0.780 | 0.835 | **0.902** |
| **HWU64** | GPT-3.5-turbo | 0.260 | 0.360 | 0.280 | 0.340 | 0.280 | 0.560 | 0.360 | 0.100 | 0.260 | 0.520 | 0.850 | **0.914** |
| | Doubao-lite-4k | 0.500 | 0.500 | 0.500 | 0.480 | 0.500 | 0.520 | 0.340 | 0.360 | 0.420 | 0.700 | 0.765 | **0.873** |
| | Doubao-pro-4k | 0.640 | 0.760 | 0.620 | 0.800 | 0.640 | 0.680 | 0.600 | 0.620 | 0.640 | **1.000** | 0.862 | 0.918 |
| | Hunyuan-lite | 0.533 | 0.367 | 0.333 | 0.433 | 0.233 | 0.600 | 0.433 | 0.540 | 0.320 | 0.700 | 0.514 | **0.784** |
| | **Average** | 0.483 | 0.497 | 0.433 | 0.513 | 0.413 | 0.590 | 0.433 | 0.405 | 0.410 | 0.730 | 0.748 | **0.872** |
| **LIU54** | GPT-3.5-turbo | 0.380 | 0.260 | 0.360 | 0.460 | 0.240 | 0.480 | 0.480 | 0.140 | 0.180 | 0.300 | 0.743 | **0.868** |
| | Doubao-lite-4k | 0.500 | 0.400 | 0.500 | 0.540 | 0.660 | 0.600 | 0.440 | 0.520 | 0.520 | 0.600 | 0.690 | **0.841** |
| | Doubao-pro-4k | 0.400 | 0.420 | 0.400 | 0.520 | 0.520 | 0.800 | 0.760 | 0.500 | 0.520 | **0.900** | 0.829 | 0.884 |
| | Hunyuan-lite | 0.533 | 0.500 | 0.567 | 0.700 | 0.633 | 0.367 | 0.500 | 0.460 | 0.620 | 0.560 | 0.565 | **0.704** |
| | **Average** | 0.453 | 0.395 | 0.457 | 0.555 | 0.513 | 0.562 | 0.545 | 0.405 | 0.460 | 0.590 | 0.707 | **0.824** |

lite, as well as open-source alternatives including Gemma, LLaMA, and Qwen models. Each dataset is used to test the models' accuracy in understanding and classifying various domain-specific tasks. The results are presented in tables, with the top-performing techniques highlighted in **bold** and the second-best results underlined for clarity.

### 4.4.1. ANALYSIS OF REASONING PERFORMANCE ON CLOSED-SOURCE MODELS

As illustrated in Table 1, RDES/B and RDES/C consistently outperform alternative methodologies across the evaluated datasets, with RDES/C, which incorporates CoT reasoning, achieving the highest accuracy scores in nearly all instances. While no single prompt engineering method dominated across all datasets, both KP and CoT reasoning yielded strong results, particularly on the HWU64 dataset, where CoT achieved the highest average accuracy of 0.513. Task-specific prompt engineering appears crucial, though its advantages are often dataset-dependent. In terms of demonstration selection, ADA and FSC produced competitive results, especially on the CLINC150 and HWU64 datasets, with ADA showing flexibility in curating demonstrations based on task-specific factors. However, it was generally outperformed by RDES/B and RDES/C, indicating that the integration of CoT reasoning in RDES/C provides significant benefits. Among the evaluated models, Doubao-pro-4k excelled, achieving a peak performance score of 0.961 on the CLINC150 dataset with RDES/C, while Doubao-lite-4k struggled, particularly on challenging datasets like HWU64, highlighting the importance of model capacity and architecture. GPT-3.5-turbo also demonstrated stable per-

formance across various datasets, reinforcing its versatility when paired with advanced techniques like RDES/C. The comparison between RDES/B and RDES/C reveals that the latter's incorporation of CoT reasoning consistently leads to superior performance, as seen in the BANKING77 dataset with an average accuracy of 0.838, significantly surpassing traditional methods like SF and ADA. This trend is echoed across other datasets, with RDES/C achieving an average accuracy of 0.902 in CLINC150, further emphasizing its effectiveness. Overall, the evaluation underscores the substantial impact of advanced prompt engineering and demonstration selection techniques on the performance of closed-source LLMs, suggesting that adaptive, context-aware prompting strategies, particularly those integrating CoT reasoning, are essential for optimizing LLMs for domain-specific tasks and guiding future research in this area.

### 4.4.2. ANALYSIS OF REASONING PERFORMANCE ON OPEN-SOURCE MODELS

Table 2 reveals significant performance variations across different datasets, highlighting the unique challenges faced by open-source LLMs. Our RDES/C approach consistently outperforms other methods in the BANKING77 dataset, demonstrating its effectiveness in understanding fine-grained customer service intents, and shows similar robustness in the HWU64 dataset across diverse user queries. The CLINC150 dataset, characterized by its technical nature, benefits notably from our methods, particularly when utilizing larger models like Qwen-1.5-72B, underscoring the importance of scale in managing domain-specific content. In the LIU54 dataset, which features specialized queries, both RDES/B

*Table 2.* Performance comparison of methods designed to boost LLM reasoning across various datasets on open-source LLMs, with a focus on accuracy. The RDES/B denotes the base version, while RDES/C indicates the version enhanced with CoT reasoning.

| Datasets | Models | Prompt Engineering Methods | | | | | Demonstration Selection Methods | | | | | Ours | |
|---|---|---|---|---|---|---|---|---|---|---|---|---|---|
| | | ZS | KP | L2M | CoT | SF | FS | FSC | AES | RDS | ADA | RDES/B | RDES/C |
| BANKING77 | Gemma-2-2B | 0.200 | 0.280 | 0.200 | 0.200 | 0.260 | 0.300 | 0.340 | 0.280 | 0.220 | **0.900** | 0.831 | 0.861 |
| | Gemma-2-9B | 0.560 | 0.400 | 0.500 | 0.500 | 0.400 | 0.440 | 0.380 | 0.400 | 0.400 | 0.700 | 0.831 | **0.886** |
| | LLaMA-3.2-1B | 0.120 | 0.100 | 0.000 | 0.000 | 0.100 | 0.000 | 0.000 | 0.020 | 0.040 | 0.680 | 0.024 | **0.744** |
| | LLaMA-3.2-3B | 0.200 | 0.200 | 0.400 | 0.500 | 0.300 | 0.320 | 0.060 | 0.360 | 0.440 | 0.700 | 0.770 | **0.805** |
| | LLaMA-3-8B | 0.578 | 0.560 | 0.563 | 0.552 | 0.458 | 0.090 | 0.182 | 0.531 | 0.536 | 0.758 | 0.784 | **0.847** |
| | Qwen-2.5-7B | 0.700 | 0.480 | 0.600 | 0.420 | 0.480 | 0.440 | 0.180 | 0.440 | 0.420 | 0.700 | 0.803 | **0.859** |
| | Qwen-2.5-14B | 0.400 | 0.400 | 0.420 | 0.420 | 0.420 | 0.480 | 0.460 | 0.500 | 0.520 | 0.800 | 0.839 | **0.868** |
| | Qwen-1.5-72B | 0.529 | 0.480 | 0.524 | 0.528 | 0.551 | 0.653 | 0.612 | 0.509 | 0.542 | 0.775 | 0.785 | **0.892** |
| | **Average** | 0.411 | 0.363 | 0.401 | 0.390 | 0.371 | 0.340 | 0.277 | 0.380 | 0.390 | 0.752 | 0.708 | **0.845** |
| CLINC150 | Gemma-2-2B | 0.400 | 0.600 | 0.420 | 0.460 | 0.560 | 0.560 | 0.540 | 0.500 | 0.380 | 0.800 | 0.875 | **0.929** |
| | Gemma-2-9B | 0.700 | 0.700 | 0.800 | 0.800 | 0.700 | 0.800 | 0.680 | 0.800 | 0.800 | 0.780 | **0.864** | 0.819 |
| | LLaMA-3.2-1B | 0.400 | 0.520 | 0.060 | 0.380 | **0.600** | 0.000 | 0.020 | 0.080 | 0.080 | 0.400 | 0.256 | 0.122 |
| | LLaMA-3.2-3B | 0.800 | 0.700 | 0.800 | 0.400 | 0.700 | 0.580 | 0.260 | 0.600 | 0.680 | 0.800 | **0.845** | 0.703 |
| | LLaMA3-8B | 0.523 | 0.439 | 0.594 | 0.504 | 0.569 | 0.007 | 0.285 | 0.571 | 0.543 | 0.767 | **0.840** | 0.783 |
| | Qwen-2.5-7B | 0.740 | 0.800 | 0.800 | 0.780 | 0.700 | 0.740 | 0.460 | 0.780 | 0.780 | 0.800 | **0.879** | 0.741 |
| | Qwen-2.5-14B | 0.900 | 0.900 | 0.900 | 0.800 | 0.840 | 0.700 | 0.700 | 0.740 | 0.740 | 0.900 | **0.944** | 0.792 |
| | Qwen-1.5-72B | 0.726 | 0.517 | 0.683 | 0.641 | 0.660 | 0.850 | 0.652 | 0.696 | 0.656 | 0.861 | 0.897 | **0.963** |
| | **Average** | 0.649 | 0.647 | 0.632 | 0.596 | 0.666 | 0.530 | 0.450 | 0.596 | 0.582 | 0.763 | **0.800** | 0.731 |
| HWU64 | Gemma2-2B | 0.300 | 0.320 | 0.300 | 0.300 | 0.400 | 0.460 | 0.440 | 0.420 | 0.360 | 0.600 | 0.832 | **0.851** |
| | Gemma2-9B | 0.600 | 0.600 | 0.600 | 0.600 | 0.600 | 0.700 | 0.700 | 0.700 | 0.700 | 0.800 | 0.877 | **0.910** |
| | LLaMA-3.2-1B | 0.200 | 0.100 | 0.080 | 0.000 | 0.100 | 0.020 | 0.000 | 0.020 | 0.060 | 0.360 | 0.381 | **0.687** |
| | LLaMA-3.2-3B | 0.300 | 0.100 | 0.300 | 0.200 | 0.300 | 0.220 | 0.180 | 0.300 | 0.300 | 0.700 | 0.747 | **0.817** |
| | LLaMA-3-8B | 0.478 | 0.407 | 0.493 | 0.479 | 0.563 | 0.632 | 0.498 | 0.651 | 0.645 | 0.837 | 0.816 | **0.859** |
| | Qwen-2.5-7B | 0.780 | 0.700 | 0.800 | 0.600 | 0.800 | 0.640 | 0.540 | 0.760 | 0.740 | 0.800 | 0.805 | **0.880** |
| | Qwen-2.5-14B | 0.780 | 0.800 | 0.800 | 0.440 | 0.740 | 0.800 | 0.800 | 0.720 | 0.680 | **0.900** | 0.886 | 0.895 |
| | Qwen-1.5-72B | 0.698 | 0.615 | 0.676 | 0.661 | 0.668 | 0.825 | 0.817 | 0.749 | 0.774 | 0.877 | 0.867 | **0.924** |
| | **Average** | 0.517 | 0.455 | 0.506 | 0.410 | 0.521 | 0.537 | 0.497 | 0.540 | 0.532 | 0.734 | 0.776 | **0.853** |
| LIU54 | Gemma-2-2B | 0.400 | 0.400 | 0.500 | 0.400 | 0.400 | 0.620 | 0.480 | 0.500 | 0.440 | 0.600 | 0.733 | **0.854** |
| | Gemma-2-9B | 0.500 | 0.500 | 0.600 | 0.600 | 0.600 | 0.580 | 0.580 | 0.500 | 0.500 | **1.000** | 0.722 | 0.837 |
| | LLaMA-3.2-1B | 0.200 | 0.160 | 0.300 | 0.400 | 0.080 | 0.040 | 0.040 | 0.360 | 0.320 | **0.700** | 0.058 | 0.651 |
| | LLaMA-3.2-3B | 0.400 | 0.400 | 0.400 | 0.300 | 0.400 | 0.360 | 0.320 | 0.500 | 0.400 | 0.600 | **0.772** | 0.749 |
| | LLaMA-3-8B | 0.358 | 0.409 | 0.428 | 0.360 | 0.392 | 0.396 | 0.320 | 0.347 | 0.312 | 0.763 | 0.779 | **0.811** |
| | Qwen-2.5-7B | **0.800** | 0.700 | 0.700 | 0.640 | 0.520 | 0.620 | 0.500 | 0.500 | 0.660 | 0.800 | 0.794 | 0.765 |
| | Qwen-2.5-14B | 0.700 | 0.860 | 0.700 | 0.660 | 0.700 | 0.660 | 0.600 | 0.740 | 0.780 | **1.000** | 0.849 | 0.743 |
| | Qwen-1.5-72B | 0.496 | 0.445 | 0.487 | 0.491 | 0.550 | 0.609 | 0.647 | 0.514 | 0.492 | 0.769 | 0.781 | **0.880** |
| | **Average** | 0.482 | 0.484 | 0.514 | 0.481 | 0.455 | 0.486 | 0.436 | 0.495 | 0.488 | 0.779 | 0.686 | **0.786** |

and RDES/C exhibit considerable advantages, showcasing their capacity for nuanced reasoning. Methodologically, the comparison highlights the strengths and weaknesses of various prompt engineering and demonstration selection techniques, with ZS and KP showing limitations compared to ADA and our RDES methods. While demonstration selection strategies like FS and FSC provide moderate improvements, they are often outpaced by more sophisticated approaches like AES and ADA, validating the efficacy of CoT reasoning in enhancing model understanding, particularly for complex tasks.

The analysis reveals distinct performance patterns tied to model architecture and scale. Smaller models, such as Gemma-2-2B and LLaMA-3.2-1B, generally exhibit lower accuracy, particularly with simpler prompting strategies, indicating limited capacity for nuanced comprehension. In contrast, larger models like Qwen-2.5-14B and Qwen-1.5-72B show marked performance improvements, especially

when combined with advanced methods like RDES/C, highlighting the synergistic effect of scale and sophisticated reasoning techniques. Overall, our findings emphasize that RDES methods, particularly when integrated with CoT reasoning, provide a clear advantage across all datasets, reinforcing the necessity of computational resources for achieving higher accuracy. The dataset-specific trends further underscore the importance of tailored approaches, as different datasets require distinct handling strategies for optimal results. This study suggests that future research should focus on refining adaptive and CoT-based methods to enhance model generalization across domains, positioning adaptive reasoning as a core component for advancing LLM architectures and increasing their versatility in real-world applications.

### 4.4.3. AVERAGE PERFORMANCE OF CLOSED-SOURCE AND OPEN-SOURCE MODELS

The data presented in Figure 3 summarizes the average performance results of various LLMs, encompassing both closed-source and open-source variants, across different datasets. This figure highlights the effectiveness of prompt engineering and demonstration selection methods in comparison to our proposed approaches, RDES/B and RDES/C. In the BANKING77 dataset, RDES/C achieved a remarkable accuracy of 0.843, significantly surpassing other methodologies, including RDES/B at 0.718 and ADA at 0.689, while the baseline method, ZS, recorded the lowest accuracy of 0.394. In the CLINC150 dataset, RDES/B demonstrated strong performance with an accuracy of 0.812, followed closely by RDES/C at 0.788, with ADA achieving 0.769 and the KP method at 0.640. The HWU64 dataset further highlighted RDES/C's superiority, as it led with an accuracy of 0.859, while RDES/B achieved 0.767 and ADA recorded 0.733, with ZS lagging at 0.506. Finally, in the LIU54 dataset, RDES/C attained an accuracy of 0.799, outperforming RDES/B at 0.693 and ADA at 0.716, while the CoT method exhibited an accuracy of 0.506. Overall, these results illustrate the effectiveness of our proposed approaches in enhancing model performance through advanced prompt engineering and demonstration selection strategies, underscoring their potential to improve classification accuracy across various applications.

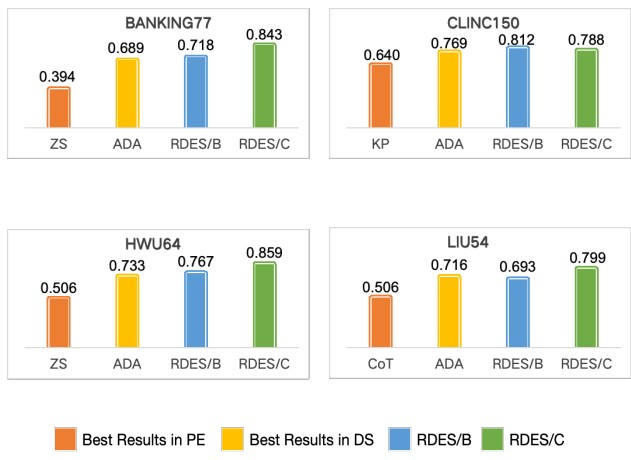

*Figure 3.* These figures illustrate the average results across closed-source/open-source models on different datasets, comparing the best results from the prompt engineering (PE) and demonstration selection (DS) methods with our proposed approach.

### 4.5. Evaluation on More Challenging Reasoning Tasks

To further assess the generalizability and effectiveness of RDES beyond simple text classification, we conducted additional experiments on more challenging reasoning benchmarks: BigBenchHard (specifically, boolean expressions

and web of lies subsets), SST5 and GSM-8K. These datasets require advanced reasoning capabilities from the LLMs, allowing us to evaluate how different demonstration selection methods perform in complex scenarios. We evaluated the performance of our proposed RDES variants (RDES/B and RDES/C based on Q-learning, and RDES/PPO based on PPO) against several baseline methods (FS, FSC, AES, RDS, ADA) on these benchmarks, utilizing both the DeepSeek-R1-32B and Qwen-2.5-72B models.

As shown in Table 5 in Appendix A.6, the results on these reasoning benchmarks further highlight the competitive performance of RDES, especially RDES/C which incorporates CoT reasoning, and RDES/PPO. For instance, RDES/C achieves the highest accuracy on both BigBenchHard and GSM-8K with the DeepSeek-R1-32B model. With the Qwen-2.5-72B model, ADA shows the highest accuracy on these specific reasoning tasks, while RDES/PPO also achieves competitive results. These findings demonstrate that RDES, by effectively balancing relevance and diversity in demonstration selection through a RL framework, can maintain strong performance even on tasks requiring more complex reasoning, supporting its broader applicability beyond straightforward classification. The exploration of different RL algorithms like PPO also shows promise for these tasks.

## 5. Conclusion

In this study, we introduced RDES, a novel framework utilizing RL (specifically Q-learning, with exploration into a PPO-based variant) to optimize demonstration selection for ICL in LLMs by balancing relevance and diversity, thereby enhancing generalization and mitigating overfitting. Our extensive evaluation against ten baselines on four benchmark classification datasets demonstrated that RDES significantly outperforms existing methods. We also showed that integrating RDES with CoT reasoning (RDES/C) generally enhances performance, though its benefit can vary depending on model and dataset characteristics. We conducted additional experiments on more challenging reasoning benchmarks and with a variable number of demonstrations, which further validated RDES's effectiveness and highlighted diversity-driven generalization, especially with the RDES/PPO variant, even in complex tasks or varying settings. These results underscore the potential of RL to facilitate adaptive demonstration selection and its promise for addressing complexities in NLP tasks. Future work includes refining diversity metrics, extending RDES to tasks beyond classification like generation and question answering, making CoT usage adaptive within the RL framework, analyzing computational cost/sample efficiency, exploring different retrieval methods, and assessing the generalization capabilities of strategies across datasets.

## Acknowledgements

This work was supported in part by the Chinese National Research Fund (NSFC) under Grant 62272050 and Grant 62302048; in part by the Guangdong Key Lab of AI and Multi-modal Data Processing, United International College (UIC), Zhuhai under 2023-2024 Grants sponsored by Guangdong Provincial Department of Education; in part by Institute of Artificial Intelligence and Future Networks (BNU-Zhuhai) and Engineering Center of AI and Future Education, Guangdong Provincial Department of Science and Technology, China; Zhuhai Science-Tech Innovation Bureau under Grant No. 2320004002772, and in part by the Interdisciplinary Intelligence SuperComputer Center of Beijing Normal University (Zhuhai).

## Impact Statement

This paper introduces the RDES framework, leveraging RL to optimize demonstration selection for LLMs in few-shot text classification and reasoning tasks by balancing relevance and diversity. The primary positive impact is a significant enhancement in LLM accuracy and robustness in data-limited scenarios, making them more effective for practical applications like intent detection and sentiment analysis. This approach helps mitigate the overfitting biases often associated with purely similarity-based selection. However, potential negative impacts include the risk that enhanced classification capabilities could be misused for applications like surveillance or censorship. While not the focus, extending the core method to generative tasks could, in the future, potentially contribute to the spread of misinformation. Furthermore, training RDES involves significant computational cost due to numerous LLM calls, potentially limiting its accessibility. The current work lacks user studies, meaning real-world human-centric impacts like user over-reliance are not yet evaluated. To maximize positive impacts and mitigate risks, future work should explore computational efficiency improvements, necessitate safeguards against misuse, particularly if extended to more complex tasks like generation, and conduct user studies to comprehensively assess real-world performance and user interaction. Adherence to strong ethical principles regarding transparency, fairness, and accountability is encouraged in the application and dissemination of this technology.

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

# A. Appendix

## A.1. Algorithm Comparison between Q-learning and PPO

In this section, we provide a comparative analysis of two prominent RL algorithms: Q-learning and Proximal Policy Optimization (PPO). The table below highlights key aspects of each algorithm, including their policy types, exploration strategies, value estimation methods, and update rules. This comparison aims to elucidate the strengths and weaknesses of each approach in the context of RL applications.

*Table 3.* Algorithm Comparison

| Aspect | Q-learning | PPO |
|---|---|---|
| Policy Type | Deterministic | Stochastic |
| Exploration | $\epsilon$-greedy | Entropy bonus |
| Value Est. | Tabular | Neural network |
| Update Rule | Temporal Difference | Surrogate objective |

## A.2. Convergence Analysis

Theoretical analysis supports the convergence properties of the adopted RL algorithms under standard conditions.

### A.2.1. Q-LEARNING CONVERGENCE

Under standard stochastic approximation conditions for the learning rates and bounded rewards, the Q-learning updates are guaranteed to converge.

**Theorem A.1** (Q-Learning Convergence). *Given learning rates $\alpha_t$ satisfying the Robbins-Monro conditions ($\sum_{t=1}^{\infty} \alpha_t = \infty$ and $\sum_{t=1}^{\infty} \alpha_t^2 < \infty$), and bounded rewards $|r_t| \leq R_{\max}$, the Q-learning updates converge almost surely to the optimal Q-function $Q^*$.*

### A.2.2. PPO POLICY IMPROVEMENT

The PPO algorithm's clipped objective provides theoretical guarantees on policy improvement in each update step.

**Theorem A.2** (Policy Improvement). *For any policy $\pi_{\theta_{old}}$, a policy $\pi_{\theta_{new}}$ updated via PPO's clipped objective with advantage estimates $\hat{A}_t$ satisfies:*

$$\mathbb{E}_{\pi_{\theta_{new}}}[\hat{A}_t] \geq \mathbb{E}_{\pi_{\theta_{old}}}[\hat{A}_t] \tag{12}$$

*This means the expected advantage of the new policy is greater than or equal to that of the old policy, guaranteeing monotonic policy improvement unless the policy has already converged.*

## A.3. Datasets

In this section, we present a summary of the datasets used in our experiments. Each dataset is characterized by its unique intents, the size of the knowledge base (KB), the number of test samples, and the distinct domains it covers. Table 4 provides a comprehensive overview of these datasets:

| Dataset | Intents | Size of KB | Test | Domains |
|---|---|---|---|---|
| BANKING77 | 77 | 9,003 | 3,080 | 1 |
| CLINC150 | 150 | 18,000 | 2,250 | 10 |
| HWU64 | 64 | 8,828 | 1,104 | 21 |
| LIU54 | 54 | 20,382 | 2,548 | 21 |

*Table 4.* Summary of Experimental Datasets: This table presents the number of unique intents represented in each dataset (**Intents**), the total number of knowledge base entries available for retrieval (**Size of KB**), the number of test samples used for evaluation (**Test**), and the number of distinct domains covered by each dataset (**Domains**).

In addition to the datasets summarized in Table 4, we performed supplementary experiments on several reasoning benchmarks

to evaluate RDES performance on tasks requiring more complex capabilities. These datasets, including BigBenchHard (Suzgun et al., 2023) (boolean expressions and web of lies subsets), GSM-8K (Cobbe et al., 2021), and SST5 (Socher et al., 2013), were utilized to address reviewer feedback and demonstrate the framework's applicability beyond standard text classification. Due to time constraints, we randomly sampled 1,000 examples from the test sets for evaluation. The experimental results and analysis on these benchmarks are presented and discussed in the main body.

### A.4. Baselines

In this study, we conduct a comprehensive evaluation of ten baseline approaches to assess their effectiveness in various classification tasks. These approaches are categorized into two main groups: Prompt Engineering Methods and Demonstration Selection Methods. Each method employs distinct strategies to enhance the model's performance, leveraging different aspects of prompting and demonstration selection to improve classification accuracy and response quality.

#### A.4.1. PROMPT ENGINEERING METHODS

This category focuses on techniques that manipulate the structure of prompts to guide the model's understanding and decision-making process. The methods included in this category are:

- **Zero-Shot Prompting (ZS)** (Radford et al., 2019): This method prompts the model to classify text without providing prior demonstrations, directly asking it to select the most appropriate label from a predefined set of options. This approach tests the model's ability to generalize from its training data.

- **Knowledge Prompting (KP)** (Liu et al., 2022): This technique prompts the model to generate relevant contextual information about the input text before selecting a label. By providing additional context, this method may enhance classification accuracy and improve the model's understanding of the task.

- **Least-to-Most Prompting (L2M)** (Zhou et al., 2023): This approach structures the classification task into smaller, manageable steps, guiding the model through the selection process. By breaking down the task, the model can focus on each component, potentially leading to more accurate predictions.

- **Chain of Thought (CoT)** (Wei et al., 2022): This variant encourages the model to articulate its reasoning step-by-step before arriving at a classification. By making the reasoning process explicit, CoT prompting can enhance response quality and provide insights into the model's decision-making.

- **Self-Refine (SF)** (Madaan et al., 2024): This approach prompts the model to solve a problem, critique its own solution, and then refine its answer based on the critique. This iterative process continues until a stopping condition is met, allowing the model to improve its responses through self-assessment.

#### A.4.2. DEMONSTRATION SELECTION METHODS

This category encompasses methods that utilize a selection of demonstrations to inform the model's predictions. The approaches included in this category are:

- **Few-Shot Prompting (FS)** (Lu et al., 2024): This approach provides the model with a limited number of text-label pairs selected randomly. By leveraging contextual information from these demonstrations, the model can enhance its predictions based on the provided demonstrations.

- **Few-Shot with CoT (FSC)** (Lu et al., 2024): This method combines few-shot learning with reasoning, presenting demonstrations alongside explanations to guide the model's classification process. This integration aims to improve the model's understanding and accuracy in making predictions.

- **Active Demonstration Selection (AES)** (Zhang et al., 2022): This approach involves iteratively selecting and annotating unlabeled demonstrations to enhance ICL using RL methods. By actively choosing relevant demonstrations, the model can improve its learning efficiency.

- **Representative Demonstration Selection (RDS)** (Yang et al., 2023): This method aims to identify a high-quality and diverse subset of in-context demonstrations that can effectively prompt various test instances for a specific task. By ensuring diversity, this approach enhances the model's ability to generalize across different scenarios.

- **Adaptive Demonstration Selection for In-Context Learning (ADAICL, ADA)** (Mavromatis et al., 2023): This approach employs a model-adaptive, optimization-free algorithm to identify uncertain demonstrations and perform semantic diversity-based selection. By focusing on uncertain cases, the model can improve its robustness and adaptability.

## A.5. LLMs Used in Experiments

To evaluate the effectiveness of the proposed method, we employ a diverse range of LLMs, encompassing both closed-source and open-source options. Closed-source models, such as GPT-3.5-turbo, Doubao-lite-4k, Doubao-pro-4k, and Hunyuan-lite, are proprietary systems developed by leading technology companies. For instance, GPT-3.5-turbo, created by OpenAI, is renowned for its advanced NLP capabilities, making it a popular choice for various applications, including conversational agents and content generation (Zhang et al., 2024). Similarly, Doubao-pro, released by ByteDance in May 2024, excels in multiple benchmarks, demonstrating strong performance in natural language understanding and generation tasks, positioning it as a versatile tool for applications ranging from question answering to complex text creation (Team, 2024a). Hunyuan-lite, developed by Tencent, is distinguished by its extensive parameter count and advanced capabilities in handling long-context inputs, thereby enhancing its performance across diverse tasks (Cloud, 2024).

In contrast, open-source models such as Gemma-2-2B, Gemma-2-9B, LLaMA-3.2-1B, LLaMA-3.2-3B, LLaMA-3-8B, Qwen-2.5-7B, Qwen-2.5-14B, and Qwen-1.5-72B offer researchers and developers the flexibility to modify and adapt the models for specific use cases. The Gemma series emphasizes efficient training techniques while maintaining high performance across various NLP tasks, encouraging customization and experimentation within the community (Team et al., 2024). The Qwen series, particularly noted for its scalability and adaptability, allows users to fine-tune models according to their needs, fostering collaboration and innovation in AI research (Bai et al., 2023; Yang et al., 2024). The LLaMA series has garnered attention for its performance across various benchmarks while enabling users to tailor the models to their specific requirements (Dubey et al., 2024). The open-source nature of these models promotes collaboration and innovation within the AI community, facilitating a broader range of experiments and applications compared to their closed-source counterparts.

In addition to these models, for supplementary experiments conducted on challenging reasoning benchmarks (such as BigBenchHard, GSM-8K, and SST5) as detailed in Section 4.5, we also utilized the Qwen-2.5-72B (Team, 2024b) and DeepSeek-R1-32B (DeepSeek-AI, 2025) models. These additional experiments were performed to demonstrate RDES's performance in more complex scenarios and were specifically included in response to reviewer feedback. The detailed results and analysis using these models are presented in the main body of the paper.

## A.6. Supplementary Results

Table 5 presents a performance comparison of the Qwen-2.5-72B and DeepSeek-R1-32B models across several supplementary datasets, including SST5, BigBenchHard (specifically focusing on boolean expressions and the web of lies tasks), and GSM-8K. It lists various methods, encompassing traditional Few-Shot (FS) and Few-Shot with Chain of Thought (FSC) approaches, as well as demonstration selection methods such as Active Demonstration Selection (AES), Representative Demonstration Selection (RDS), and Adaptive ICL (ADA). Notably, the table also includes different variants of the proposed RL-based RDES method: RDES/B, RDES/C (the base version and the one combined with Chain of Thought), and RDES/PPO (the PPO-based variant). These supplementary results further support the effectiveness of the RDES method across diverse tasks and models, including text classification, complex reasoning, and mathematical reasoning.

*Table 5.* Supplementary Performance Comparison on SST5, BigBenchHard, and GSM-8K

| Methods | SST5 | | BigBenchHard - boolean expressions | | BigBenchHard - web of lie | | GSM-8K | |
|---|---|---|---|---|---|---|---|---|
| | Qwen-2.5-72B | DeepSeek-R1-32B | Qwen-2.5-72B | DeepSeek-R1-32B | Qwen-2.5-72B | DeepSeek-R1-32B | Qwen-2.5-72B | DeepSeek-R1-32B |
| FS | 0.56 | 0.70 | 0.98 | 0.38 | 0.58 | 0.98 | 0.50 | 0.28 |
| FSC | 0.54 | 0.66 | 0.60 | 0.46 | 1.00 | 1.00 | 0.56 | 0.64 |
| AES | 0.84 | 0.84 | 0.53 | 0.60 | 0.85 | 0.72 | 0.92 | 0.08 |
| RDS | 0.76 | 0.84 | 0.53 | 0.60 | 0.89 | 0.68 | 0.90 | 0.48 |
| ADA | 0.90 | 0.90 | 0.53 | 0.60 | 0.83 | 0.72 | 0.98 | 0.36 |
| RDES/B | 0.44 | 0.57 | 0.76 | 1.00 | 0.50 | 0.93 | 0.87 | 0.37 |
| RDES/C | 0.51 | 0.52 | 0.90 | 0.99 | 0.98 | 1.00 | 0.92 | 0.73 |
| RDES/PPO | 0.84 | 0.84 | 1.00 | 1.00 | 1.00 | 0.90 | 0.94 | 0.48 |

Table 6 presents experimental results on the GSM-8K and SST5 datasets, specifically investigating the impact of varying the number of demonstrations (k) used for ICL. The results are reported for the Qwen-72B model. The table evaluates

the performance (Accuracy) of various demonstration selection methods, including FS, FSC, AES, RDS, ADA, RDES/B, RDES/C, and RDES/PPO, as the number of demonstrations is set to 3, 5, 7, and 10. These results, included as part of the authors' experimental revisions, highlight how the performance of different methods can change depending on the size of the demonstration set.

*Table 6.* Performance of Methods Across Varying Numbers of Demonstrations (k) Using Qwen-2.5-72B Model

| Methods | GSM-8K (Accuracy) | | | | SST5 (Accuracy) | | | |
|---|---|---|---|---|---|---|---|---|
| | k=3 | k=5 | k=7 | k=10 | k=3 | k=5 | k=7 | k=10 |
| FS | 0.50 | 0.28 | 0.50 | 0.28 | 0.54 | 0.56 | 0.54 | 0.56 |
| FSC | 0.56 | 0.64 | 0.56 | 0.64 | 0.52 | 0.54 | 0.52 | 0.54 |
| AES | 0.92 | 0.08 | 0.92 | 0.08 | 0.82 | 0.84 | 0.82 | 0.84 |
| RDS | 0.90 | 0.48 | 0.90 | 0.48 | 0.74 | 0.76 | 0.74 | 0.76 |
| ADA | 0.98 | 0.36 | 0.98 | 0.36 | 0.88 | 0.90 | 0.88 | 0.90 |
| RDES/B | 0.87 | 0.37 | 0.87 | 0.37 | 0.42 | 0.44 | 0.42 | 0.44 |
| RDES/C | 0.92 | 0.73 | 0.92 | 0.73 | 0.49 | 0.51 | 0.49 | 0.51 |
| RDES/PPO | 0.94 | 0.48 | 0.94 | 0.48 | 0.82 | 0.84 | 0.82 | 0.84 |

## A.7. Ablation Study

This ablation study investigates the impact of diversity mechanisms—namely, No-Diversity, RDES/B, and RDES/C—across both closed-source and open-source models using four datasets. The results, illustrated in Figures 4 and 5, consistently demonstrate that incorporating diversity generally enhances model performance, although the degree of improvement varies depending on the dataset and model type.

In the closed-source context (Figure 4), the RDES/C mechanism consistently yields superior results across all datasets. In the BANKING77 dataset, RDES/C achieves an average accuracy of 0.838, significantly higher than the No-Diversity baseline of 0.600, indicating its effectiveness in handling diverse banking-related queries. Similarly, in the CLINC150 dataset, RDES/C reaches 0.902 in accuracy, outperforming the No-Diversity baseline of 0.770, demonstrating improved capacity to interpret a broad range of user intents. The HWU64 dataset also shows a clear benefit from diversity mechanisms, with RDES/C achieving 0.872 in accuracy, up from 0.690 in the No-Diversity setting, highlighting the mechanism's advantage in technical and domain-specific scenarios. In the LIU54 dataset, RDES/C's accuracy of 0.824 surpasses the 0.640 baseline of No-Diversity, further emphasizing the benefits of diversity for nuanced classification tasks. Overall, the RDES/C mechanism stands out as the most effective strategy for closed-source models, particularly in complex and varied environments.

For open-source models (Figure 5), the impact of diversity mechanisms shows more variation. In the BANKING77 dataset, RDES/C provides a modest improvement, achieving an average accuracy of 0.845 compared to the No-Diversity baseline of 0.747, suggesting moderate gains from diversity, especially in larger models such as Qwen-1.5-72B. However, in the CLINC150 dataset, RDES/B outperforms both No-Diversity (0.768) and RDES/C (0.731), reaching an average accuracy of 0.800. This suggests that the choice of diversity mechanism should be informed by dataset characteristics, as RDES/B appears more effective in certain contexts. A clearer trend is observed in the HWU64 dataset, where RDES/C significantly boosts accuracy to 0.853 from the No-Diversity baseline of 0.732, underscoring the value of diversity in handling complex queries. In the LIU54 dataset, performance differences are subtler, with No-Diversity and RDES/B yielding similar results (0.769 vs. 0.686), while RDES/C achieves a slightly higher accuracy of 0.786. These findings indicate that while diversity mechanisms often enhance performance, the specific choice and impact may vary across different datasets and open-source models, requiring a nuanced approach to model and dataset pairing.

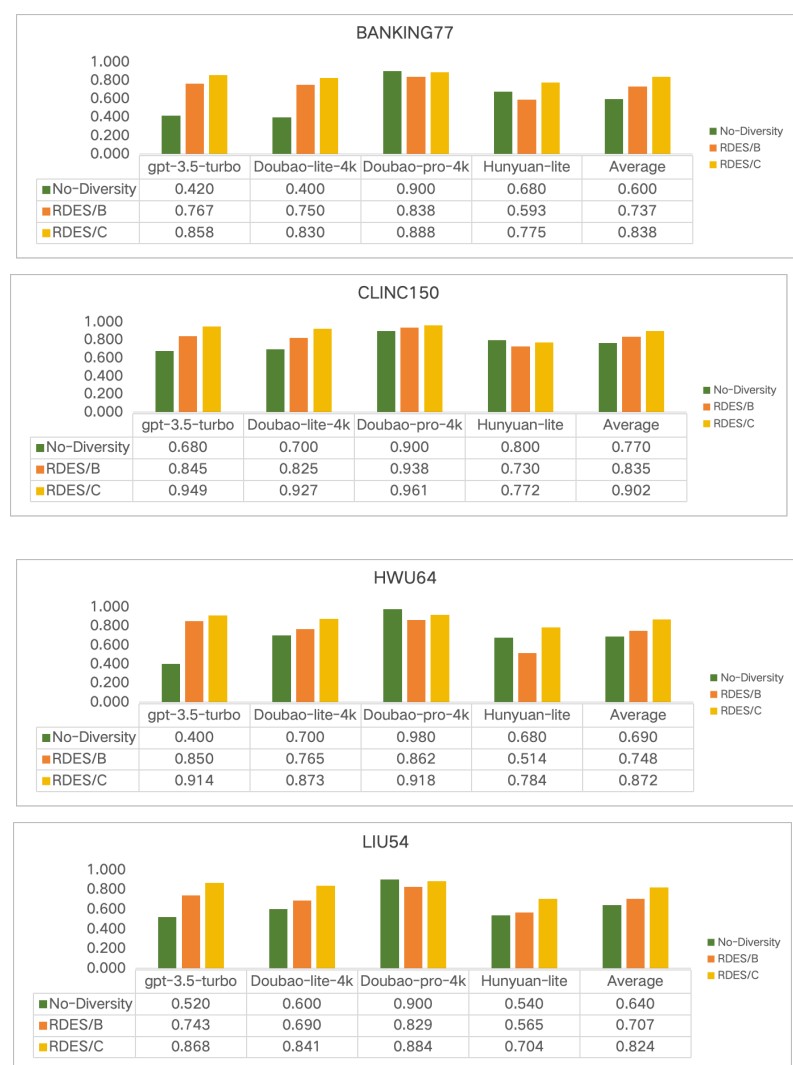

*Figure 4.* These figures illustrate the performance of various closed-source models across different datasets, highlighting the impact of diversity mechanisms.

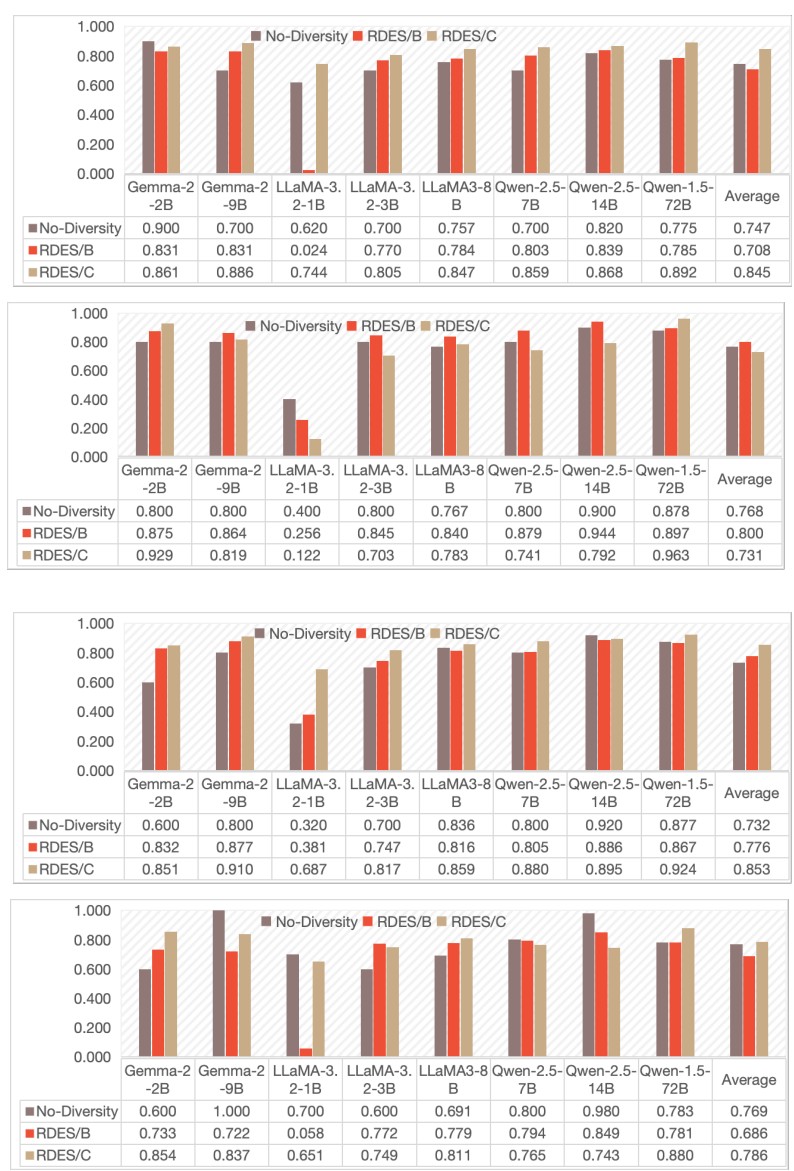

*Figure 5.* These figures illustrate the performance of various open-source models across different datasets, highlighting the impact of diversity mechanisms.

