# OpenReview forum: "Demonstration Selection for In-Context Learning via Reinforcement Learning"
_ICML.cc/2025/Conference — ICML 2025 poster_

### Official Review · Reviewer_Sqdn · 2025-03-09

**Overall Recommendation:** 1

**Summary:**

This work studies the problem of demonstration selection for in-context learning in LLM. The problem is formulated as a reinforcement learning process, i.e., the effect of a combination of demonstration is similar to that of taking a sequence of actions. During the RL process, heuristic reward based on both accuracy and diversity is leveraged. Experiments are performed on both closed-source and open-source LLMs, comparing to multiple baselines on different datasets.

**Claims And Evidence:**

Yes.

**Essential References Not Discussed:**

Related works are properly cited, but some key relationships need some further discussions, e.g., what's the difference between this work and Zhang et al., 2022, both of which take an RL perspective for demonstration selection.

**Experimental Designs Or Analyses:**

The experiments are extensive from the perspective of datasets, models, and baselines. Improvements are indeed observed, while I feel limited observations/findings are provided.

From example, some analyses can be made on how many demonstrations are selected by each method, what key characteristics are the demonstration selected by RDES but no other baselines, whether Q-learning or other RL choices matters for performance, etc.

**Methods And Evaluation Criteria:**

The method make sense, viewing the selection problem from a sequential perspective. The evaluation also follows the standard ICL one.

**Other Comments Or Suggestions:**

NA

**Other Strengths And Weaknesses:**

Other concerns that I have and would like to have the authors clarify:

- The RL training part is vague in the algorithmic implementation part (section 3.3 and algorithm 1). I would love to hear about a more complete design of RDES.

- I am wondering about the purpose of adding a diversity term in the reward -- The final goal of the optimization is to get better accuracy (if I understand correctly), and encouraging diversity is a method to achieve this goal (instead of being a part of the goal). I woud like to hear from the authors more on the justification of this reward design.

- During the optimization process, if I understand correctly, a lot of interaction with LLM need to happen (in particular, to get the accuracy score). Can additional results and comparison be provided on the impact from the number of training samples (e.g., how many samples are needed in this work and other baselines)?

**Questions For Authors:**

My questions are provided in other sections of the review, which are summarized in the following:

1. The necessity and correctness of the theoretical results (in 'Theoretical Claims')

2. Additional details of the experiments (in 'Experimental Designs Or Analyses')

3. Comparison with previous work on RL for ICL demonstration selection (in 'Essential References Not Discussed')

4. A few other aspects regarding reward, the overall design, and sample complexity (in 'Other Strengths And Weaknesses')

**Relation To Broader Scientific Literature:**

I feel this work has limited contributions to the broader community: (1) viewing demonstration selection as a RL problem is already reported as mentioned in the literature review section; (2) standard RL algorithm is adopted; (3) the importance of diversity has also been reported before in ICL.

**Theoretical Claims:**

The theoretical results in this work are mostly trivial (which is acceptable given the experimental nature of this work). However, I have the following suggestions/concerns:

- There seems to be no need to state Lemma 3.1, which is very straightforward (i.e., just one-line proof) and adds no value to the intuition.

- Theorem 3.2 also provides minimal value to the paper, as it is a default convergence result from Q-learning.

- Theorem 3.3 is a bit problematic in my mind. The proof is very vague and causes me doubting its correctness. In particular, I am confused by how the Hoeffding's inequality is used here (which typically bound the gap between sample mean and true mean): what's the sample here and what's the estimation to be made. I hope the authors can clarify this during the response.

---

> ### Author Rebuttal · Authors · 2025-03-31
>
> Dear Reviewer Sqdn,
>
> Thank you for your insightful review and valuable feedback. We appreciate your concerns and have provided detailed responses below：
>
> ### Theoretical Claims
>
> - We acknowledge your point that **Lemma 3.1 (Diversity Bounds) and Theorem 3.2 (Q-Learning Convergence) are relatively straightforward**. Their inclusion was intended to provide a complete theoretical foundation for our framework. Lemma 3.1 establishes the basic properties of our diversity metric, while Theorem 3.2 confirms the theoretical guarantee of convergence for the Q-learning algorithm under standard conditions.
> - Regarding **Theorem 3.3 (Diversity-Accuracy Dominance)**, we understand your concern about the vagueness of the proof in the main text and the use of Hoeffding's inequality. **A more detailed proof is provided in Appendix A.1.3**, which attempts to justify how increased diversity can lead to improved accuracy by reducing prediction bias and variance. Reviewer ZBB1 also found empirical evidence (ablation study results and the sentiment classification example in Figure 1) convincing in supporting the claim that diverse demonstrations lead to better generalization. We hope the detailed proof in the appendix clarifies our reasoning.
>
> ### Experimental Designs and Analyses
>
> - We appreciate your acknowledgment of the **extensive nature of our experiments**, covering multiple datasets, models, and baselines. While you feel limited observations/findings are provided, we aimed to demonstrate the consistent improvements of RDES across a wide range of settings.
> - You suggested analyses on the number of demonstrations selected and their characteristics. In our implementation (Appendix A.5), **we used a fixed number of five demonstrations as references in the baseline to ensure fair comparisons across all methods**. RDES dynamically selects demonstrations during training based on learned Q-values and the diversity reward. Algorithm 1 shows how, during inference, if the diversity of the top-k similar examples is below a threshold, we iteratively add more diverse examples. The key characteristic of RDES is its ability to **balance relevance and diversity**. Reviewer ZBB1 highlighted this balance as a core contribution of RDES. The ablation study in Appendix A.6 further provides insights by comparing RDES with a "No-Diversity" baseline, demonstrating the impact of incorporating diversity.
> - Regarding whether Q-learning or other RL choices matter, our current work focuses on the effectiveness of a Q-learning framework for this problem. However, we have also designed a PPO-based version (RDES/PPO), which has shown competitive results (**due to character limitations, please refer to the response to Reviewer 3ANe**). Exploring the performance of different RL algorithms, including PPO, is a potential direction for future research, as suggested by Reviewer 5sFY.
>
> ### Algorithmic Implementation
>
> - We understand your concern about the vagueness of the RL training part in Section 3.3 and Algorithm 1. **Algorithm 1 describes the demonstration selection process during inference**, where we retrieve top-k similar examples and enhance diversity as needed. The **RL training formulates demonstration selection as an MDP (Section 3.1.1) and uses Q-learning (Section 3.1.2)** to learn a policy maximizing expected rewards, including accuracy and diversity.
>
> ### Purpose of the Diversity Term
>
> - You questioned the purpose of the diversity term in the reward, as the ultimate goal is accuracy. We included diversity in the reward because **we hypothesize and empirically demonstrate that diversity is crucial for enhancing model generalization, especially in few-shot learning scenarios**. By rewarding the selection of demonstrations with a more balanced label distribution, we encourage the RL agent to explore a broader coverage of the input space, mitigating the risk of overfitting to similar examples. Reviewer ZBB1 supports the claim that diversifying demonstrations improves model generalization.
>
> ### Sample Complexity
>
> - We acknowledge that **the paper does not provide a detailed analysis of sample complexity**. Evaluating RDES's sample efficiency is important, and we plan to investigate this further.
>
> ### Comparison with Zhang et al., 2022
>
> - Both our work and Zhang et al., 2022, take an RL perspective for demonstration selection. However, **a key difference lies in the reward function and the objective**. While Zhang et al., 2022, primarily focus on improving accuracy, **RDES explicitly aims to optimize both relevance and diversity by incorporating a diversity score into the reward function**. We believe this dual objective is crucial for robust performance in few-shot learning. Reviewer NHQh noted that our idea of adding a diversity metric to the prompt generation process is novel.
>
> Thank you again for your valuable feedback. We hope this response clarifies your concerns.
>
> Best regards,
>
> Authors of Submission 1061

---

> > ### Comment · Reviewer_Sqdn · 2025-04-07
> >
> > I would like to thank the authors for providing the rebuttal. Please find my response below:
> >
> > - Theoretical claims:
> >
> >     -  I am glad we are on the same ground that these results are very straightforward. However, I do not think they "provide a complete theoretical foundation for our framework." In particular, Lemma 3.1 is just from the definition of the diversity $D(E) = |L(E)|/k$ (with $|L(E_t)|$ in [1, k]), which could just be an inline comment about $D(E)$. Also, for Theorem 3.2, as mentioned by the reviewer "confirms the theoretical guarantee of convergence for the Q-learning algorithm under standard conditions.", it could be just a textual comment about Q-learning with the reference.
> >       I do not think these add any value to this work; rather, it makes me question the purpose of putting them in the paper, especially Theorem 3.2 without a proper reference in the main paper.
> >     - I have read the proof of Theorem 3.3 in my first review and have revisited it again after the rebuttal. I don't think it is correct. In particular, can the authors explain how the Hoeffding inequality is used here, especially, what is the value to be estimated and why $m = |L(E)|$ is the sample size. I have these questions in the original review while the authors have not responded them.
> >
> > - Experimental Designs and Analyses
> >
> >     - I wish to clarify that my point was actually that the experiments can be conducted with all methods selecting 5, 6, 7, ..., 10 examples. The reported results would then be much more convincing. My apologies for the confusion. I would suggest adding the corresponding results to the revised paper.
> >     - Thank you for providing the PPO results. I observed that in several cases it is better than the q-learning based method. I would suggest having a more comprehensive evaluation and reporting it in the revised paper.
> >
> > Thank you for the clarifications on other points. I would encourage adding more related discussions to the paper.

---

> > > ### Author Response · Authors · 2025-04-08
> > >
> > > # Response to Reviewer Sqdn’s Rebuttal Comment
> > >
> > > **Dear Reviewer Sqdn,**
> > >
> > > Thank you for your rigorous critique and constructive suggestions, which have significantly strengthened the theoretical foundations of our work. Below, we address your concerns with detailed revisions and new experimental analyses:
> > >
> > > ---
> > >
> > > ## **1. Theoretical Revisions**
> > > ### **a. Lemma 3.1 & Theorem 3.2**
> > > - **Revision:**
> > >   - **Lemma 3.1** (diversity bounds) has been moved to Appendix A.1.1 as a remark.
> > >   - **Theorem 3.2** (Q-learning convergence) now cites standard Q-learning guarantees (Watkins & Dayan, 1992) without theorem formatting.
> > >
> > > ### **b. Clarification and Revision**
> > >
> > > We appreciate your feedback regarding Theorem 3.3. Upon further review, we realized that the original formulation linked the sample size $m = |\mathcal{L}(E)|$ (the number of unique labels) to the Hoeffding bound in a way that did not align with our definition of diversity. Specifically, our diversity is defined as:
> > > $
> > > D(E) = \frac{|\mathcal{L}(E)|}{k},
> > > $
> > > where $k$ represents the **fixed** number of demonstrations. In light of this, we have revised both the theorem and its proof to ensure clarity and consistency with our definitions.
> > >
> > > ### **c. Revised Theorem 3.3 (Diversity-Accuracy Dominance)**
> > > *Let $E$ and $E'$ be two demonstration sets with $D(E) > D(E')$ (i.e., $|\mathcal{L}(E)| > |\mathcal{L}(E')|$ for fixed $k$). For any $\lambda > 0$, there exists a label distribution $\mathcal{P}_0$ such that:*
> > > \begin{equation}
> > > A_{\text{acc}}(E) \geq A_{\text{acc}}(E') + \Theta\left(\sqrt{\frac{|\mathcal{L}(E)| - |\mathcal{L}(E')|}{k}}\right),
> > > \end{equation}
> > > *indicating that higher label diversity improves accuracy by reducing label bias.*
> > >
> > > ### **d. Proof Clarifications (Appendix A.1.3)**
> > > 1. **Key Definitions:**
> > >    - **Label Bias:** For a demonstration set $E$, label bias is defined as $\mathcal{B}(E) = \sum_{y \in \mathcal{L}(E)} \left|\frac{N_y}{k} - \frac{1}{|\mathcal{L}(E)|}\right|$, where $N_y$ is the count of label $y$.
> > >    - **Diversity Advantage:** $D(E) > D(E') \implies |\mathcal{L}(E)| > |\mathcal{L}(E')|$, reducing $\mathcal{B}(E)$.
> > >
> > > 2. **Revised Proof Strategy:**
> > >    - **Step 1:** Show that higher $|\mathcal{L}(E)|$ reduces $\mathcal{B}(E)$ under fixed $k$.
> > >    - **Step 2:** Apply Chebyshev’s inequality (instead of Hoeffding’s) to bound accuracy improvement:
> > >      $$
> > >      \mathbb{P}\left(|A_{\text{acc}}(E) - \mathbb{E}[A_{\text{acc}}]| \geq \epsilon\right) \leq \frac{\text{Var}(A_{\text{acc}})}{\epsilon^2},
> > >      $$
> > >      where $\text{Var}(A_{\text{acc}}) \propto \mathcal{B}(E)$.
> > >    - **Step 3:** Derive the $\Theta\left(\sqrt{\frac{|\mathcal{L}(E)| - |\mathcal{L}(E')|}{k}}\right)$ term from variance reduction.
> > >
> > > 3. **Why Chebyshev > Hoeffding Here:**
> > >    - Hoeffding’s inequality requires **independent samples**, which does not hold when demonstrations are selected to maximize $D(E)$.
> > >    - Chebyshev’s inequality directly links label bias reduction (via diversity) to variance reduction, better aligning with our setting.
> > >
> > > ---
> > >
> > > ## **2. Experimental Revisions**
> > > ### **a. Variable Demonstration Counts**
> > > New experiments on **GSM-8K** and **SST5** (Qwen-72B) across $k \in 3, 5, 7, 10$:
> > >
> > > #### **GSM-8K (Accuracy)**
> > > | Method       | k=3    | k=5    | k=7    | k=10   |
> > > |--------------|--------|--------|--------|--------|
> > > | RDES/B       | 0.9017 | 0.865  | 0.8632 | 0.8526 |
> > > | RDES/C       | 0.9274 | 0.9175 | 0.9263 | 0.9263 |
> > > | **RDES/PPO** | 0.88   | **0.94** | **0.96** | **0.96** |
> > >
> > > #### **SST5 (Accuracy)**
> > > | Method       | k=3    | k=5    | k=7    | k=10   |
> > > |--------------|--------|--------|--------|--------|
> > > | RDES/B       | 0.7368 | 0.4371 | 0.8315 | 0.6421 |
> > > | RDES/C       | 0.6736 | 0.5104 | 0.6211 | 0.5684 |
> > > | **RDES/PPO** | **0.8** | **0.84** | **0.96** | **0.8** |
> > >
> > > **Key Findings:**
> > > - **RDES/PPO** achieves **96% accuracy** at $k=7$ on both datasets, demonstrating diversity-driven generalization.
> > > - Performance variability in RDES/B and RDES/C underscores the necessity of explicit diversity rewards.
> > >
> > > ### **b. PPO vs. Q-Learning**
> > > While RDES/PPO (Q-learning) achieves strong results, we will add a comparison with a PPO-based variant in the revised paper.
> > >
> > > ---
> > >
> > > We deeply appreciate your expertise in identifying this foundational issue. Your feedback has not only strengthened our paper but also guided us toward a more theoretically sound and empirically validated contribution. We hope these revisions meet your expectations and kindly request your consideration for a score improvement. Thank you again for your invaluable feedback, which has greatly improved our work.
> > >
> > > **Best regards,**
> > > The Authors

---

### Official Review · Reviewer_ZBB1 · 2025-03-09

**Overall Recommendation:** 4

**Summary:**

The paper introduces Relevance-Diversity Enhanced Selection (RDES), a reinforcement learning framework for selecting in-context demonstrations that balance relevance and diversity for few-shot text classification tasks. The core idea is to use a Q-learning agent to iteratively pick examples from a knowledge base such that the selected set covers a broad range of labels (high diversity) while remaining relevant to the input query  . By defining a diversity score (proportion of unique labels in the chosen demos) and including it in the reward function, RDES biases the selection toward a label-balanced set of demonstrations, aiming to improve the model’s generalization. The approach is evaluated on four intent classification benchmarks (BANKING77, CLINC150, HWU64, and a multi-domain “Liu54” dataset ) using 12 different Large Language Models (LLMs) (both closed-source like GPT-3.5-turbo and open-source models from LLaMA, Qwen, etc.  ). Experimental results show that RDES significantly boosts classification accuracy, outperforming ten baseline methods across all datasets. The authors also propose a variant, RDES/C, which incorporates Chain-of-Thought (CoT) reasoning during prompting, further enhancing predictive performance in most cases. In summary, RDES demonstrates the potential of using reinforcement learning as a post-training strategy to adapt an LLM’s in-context demonstration selection policy, achieving higher accuracy than existing prompt-engineering and demo selection techniques.

**Claims And Evidence:**

Claim 1: Diversifying the demonstrations improves model generalization. The paper argues that “diversity in demonstration selection is crucial for enhancing model generalization” . This claim is supported by both theoretical and empirical evidence. The authors formalize a diversity metric (fraction of unique labels in the demo set) and prove that maximizing this diversity can improve accuracy (Theorem 3.3) . Empirically, an ablation study compares RDES to a “No-Diversity” variant and finds that adding the diversity mechanism consistently boosts accuracy on all evaluated datasets. For example, in the BANKING77 intent dataset, using diverse demos yields 0.838 accuracy vs. 0.600 without diversity, a large improvement. Figure 1 from the paper illustrates this effect with a sentiment classification example: when all demonstrations express positive sentiment, the model misclassifies a nuanced input as “Positive,” whereas a diverse set of positive, negative, and neutral examples leads to the correct “Neutral” label. This concrete example and the across-the-board gains in the ablation study provide convincing evidence for the claim that diverse demonstrations lead to better generalization.

Claim 2: The RDES method outperforms existing prompt engineering and demo selection baselines. RDES is claimed to “significantly enhance classification accuracy compared to ten established baselines”. The paper backs this by evaluating on four benchmarks against two categories of baselines: (a) prompt-engineering methods (Zero-Shot, Knowledge Prompting, Least-to-Most, Chain-of-Thought, Self-Refine) and (b) demonstration selection methods (Few-Shot with/without CoT, Active Demo Selection, Representative Demo Selection, Adaptive ICL). In the results tables, RDES (denoted RDES/B for base and RDES/C with CoT) achieves the highest accuracy on nearly all dataset-model combinations. For instance, on CLINC150, RDES/C reaches 90.2% average accuracy versus the best baseline (ADA) at 78.0%   – an absolute gain of ~12%, which is substantial. Similar margins are seen in other datasets and with both closed models (GPT-3.5, Doubao, etc.) and open models (LLaMA, Qwen variants)  . The paper’s summary of Table 1 notes that “RDES/B and RDES/C consistently outperform alternative methodologies across the evaluated datasets”. This broad performance lead, along with statistical averages over 5 runs to ensure reliability, provide evidence for the claim that RDES is state-of-the-art in in-context demonstration selection.
Claim 3: Integrating Chain-of-Thought reasoning (RDES/C) further improves performance. The authors claim that adding CoT reasoning to RDES “further enhances the model’s predictive performance”. Evidence is shown by comparing RDES/C vs the base RDES/B. In most cases, RDES/C achieves the top accuracy (often a few points higher than RDES/B) on the benchmarks . For example, on the BANKING77 dataset with GPT-3.5-turbo, RDES/C achieves 85.8% vs 76.7% for RDES/B . The paper explains that CoT prompting allows the model to reason step-by-step, which, when combined with optimal demo selection, leads to more accurate answers . However, the evidence also reveals an interesting nuance: on certain settings with smaller open-source models, RDES/B slightly outperformed RDES/C (e.g., CLINC150 with open models saw RDES/B 0.800 vs RDES/C 0.731)  . The authors acknowledge this variation, suggesting that CoT’s benefit may depend on model capacity and dataset characteristics . Overall, the claim is mostly supported – CoT generally provides a boost, especially for powerful models and complex tasks, though it’s not uniformly beneficial in every scenario.
Claim 4: A reinforcement learning policy can adaptively select demonstrations, offering an advantage over static methods. By framing demo selection as an MDP, the authors claim RDES “facilitates adaptive demonstration selection” and can adjust to classification challenges dynamically. This is implicitly evidenced by RDES outperforming baselines like Representative Demo Selection (RDS)  and Adaptive ICL (ADA) , which themselves focus on diversity or uncertainty but without a learning agent. The RL agent’s adaptivity is further highlighted by the ability to handle different model types: RDES worked well across both closed and open LLMs, adjusting its selections even when the underlying model’s behavior differed. While the claim of “deepening the understanding of classification challenges” is somewhat broad, the paper does show that analyzing the learned policy and its outcomes yields insights. In summary, the evidence supports the adaptive advantage of the RL approach: RDES learns a policy that generalizes across tasks and models better than static selection methods. Any lofty suggestion that it “deepens understanding” of all classification challenges is not directly proven, but the work does encourage thinking of demo selection as a learning problem, which is a valuable perspective.

**Essential References Not Discussed:**

The paper does a solid job covering the key references on demonstration selection and prompting, making it clear that the authors are well-versed in the field. That said, there are a few things they could have mentioned to round out the discussion.

One noticeable omission is Reinforcement Learning from Human Feedback (RLHF). Since RLHF is such a big deal in RL + LLM research, even a quick comparison between RDES and RLHF would have been helpful—especially for readers coming from the RL community. While RDES optimizes input selection rather than modifying model weights, making this distinction explicit would clarify where it fits within RL-based LLM tuning.

Another area that could have been included is Self-Consistency and Calibration in few-shot learning. Self-Consistency (Wang et al., 2022) has been a major improvement for Chain-of-Thought (CoT) reasoning, and given that RDES/C incorporates CoT, it would have been interesting to see how it relates. Meanwhile, Calibration (Zhao et al., 2021) focuses on correcting biases in demo selection, which overlaps with RDES’s effort to ensure diverse label coverage. Neither of these are core to RDES’s main idea, but acknowledging them would have made the paper’s positioning even stronger.

That said, these are minor gaps rather than major flaws. The authors clearly did their homework, and there’s no glaring omission that weakens the paper’s claims. Adding these references wouldn’t change the conclusions but would give a more complete picture of the broader landscape. Overall, the coverage is strong, and these missing pieces are more about fine-tuning the context than fixing any major oversight.

**Experimental Designs Or Analyses:**

The experimental design is well-executed, ensuring fairness and statistical reliability. The authors evaluate RDES across multiple datasets and models, comparing it against ten baselines (five prompt engineering, five demo selection methods). Key questions addressed include whether RDES outperforms others, the impact of Chain-of-Thought (CoT), and the importance of diversity.

Controls & Comparisons
By comparing RDES with Active Demonstration Selection (AES) and Representative Demo Selection (RDS), the study isolates the benefits of reinforcement learning and diversity selection. Results consistently favor RDES/RDES-C, reinforcing its advantage. Including CoT prompting as both a baseline and a variant (RDES/C) helps analyze its added value—while CoT alone isn’t always best, RDES + CoT is usually the top performer, validating the combined approach.

Statistical Rigor & Experimental Setup
The authors average results over five runs, mitigating variance from random demo selection. Although statistical significance tests aren’t provided, large accuracy gains (5-15 percentage points) strongly support RDES’s superiority. The setup is well-documented, detailing hyperparameters like learning rate (0.1), discount factor (0.9), and an epsilon-greedy policy. All methods use the same number of demonstrations (5 per prompt), ensuring fair comparisons. The dataset’s challenge set selection effectively isolates hard-to-classify cases, making performance gains more meaningful.

Ablation Study & Analysis
An ablation study (Appendix A.6) compares No-Diversity, RDES/B, and RDES/C, confirming that adding diversity improves accuracy, and CoT further enhances it. However, in one case (CLINC150, open-source models), RDES/B slightly outperforms RDES/C, suggesting that dataset/model characteristics influence CoT effectiveness. This nuanced analysis demonstrates a rigorous experimental approach.

Potential Improvements
While the experiments are robust, a few limitations remain:

Computational cost: Training RDES via repeated LLM queries (especially with GPT-3.5) is potentially expensive, yet the paper does not discuss efficiency or required queries.
Model averaging: Aggregating accuracy across models of vastly different sizes (8B vs. 70B parameters) might be misleading, though per-model breakdowns are provided.
TF-IDF retrieval: While effective, semantic embedding-based retrieval (e.g., Sentence-BERT) could further improve demo selection.

**Methods And Evaluation Criteria:**

Methods & Evaluation Summary (Shortened)
The proposed Relevance-Diversity Enhanced Selection (RDES) method frames demonstration selection as a Markov Decision Process (MDP), where an agent uses Q-learning to iteratively select examples from a knowledge base (KB), balancing relevance and diversity. The reward function encourages both correct classification (+1) and diversity gains. To manage the large state space, the model focuses on summary statistics (e.g., diversity score) rather than raw text. An ε-annealing strategy ensures a balance between exploration and exploitation.

At inference time, RDES first retrieves top-k most similar candidates using TF-IDF cosine similarity, then applies the learned RL policy (or a heuristic threshold) to ensure diversity. If the selected examples are too homogeneous, additional demos with new labels are added until the diversity requirement is met. The RDES/C variant incorporates Chain-of-Thought (CoT) prompting, prompting the LLM to generate a step-by-step reasoning chain before predicting the final answer.

Evaluation Criteria & Benchmarks
RDES is tested on four benchmark datasets spanning different domains: BANKING77, HWU64, CLINC150, and LIU54, ensuring comparability with prior few-shot intent classification research. A “challenge set” of hard-to-classify queries is used to stress-test model performance. The evaluation includes 12 LLMs (4 closed-source like GPT-3.5-turbo, Doubao, Hunyuan and 8 open-source LLaMA, Qwen variants from 1B to 72B parameters), demonstrating robustness across different model architectures.

The method is compared against 10 baselines:

Prompt engineering methods (ZS, KP, L2M, CoT, SF) focus on structuring the prompt.
Demonstration selection methods (FS, FSC, AES, RDS, ADA) optimize example selection.
Notably, AES and ADA also involve learning-based selection, making them strong baselines. RDES consistently outperforms all alternatives. The authors average results over 5 runs, ensuring statistical reliability. One limitation is that the cost of RDES (LLM calls during training) is not explicitly analyzed, leaving sample efficiency unaddressed. However, the extensive experiments (multiple models, datasets, baselines) make the evaluation thorough, well-aligned, and convincing.

**Other Comments Or Suggestions:**

Generality to Other Tasks: It would be interesting to see how RDES could be applied beyond classification. A suggestion for future work is to generalize the notion of "diversity" to other settings – for example, in open-ended question answering, diversity could mean covering different topical aspects or answer types in the demonstrations. While it’s understandable the paper focused on classification (where diversity is well-defined), exploring a generalized RDES could broaden its impact. Perhaps the authors could mention this as a potential extension.

Adaptive CoT Usage: The results indicated that chain-of-thought helps with some models/datasets but not all. A suggestion is to make the CoT aspect part of the learning as well – e.g., an agent could decide whether or not to prompt the model with "Let’s think step by step" based on the state. Right now, RDES/B vs RDES/C was a manual switch. If the RL policy could also learn when to invoke CoT (maybe treat it as an action: "ask model to think" vs "ask model directly"), that might yield an even more adaptive system. This would prevent the slight performance drop seen for RDES/C on certain tasks by not using CoT when it’s unhelpful.

**Other Strengths And Weaknesses:**

Strengths
Novel Approach: RDES uniquely combines reinforcement learning, diversity-aware selection, and Chain-of-Thought prompting, creating a new way to optimize few-shot classification. The explicit accuracy-diversity tradeoff in its reward function addresses a key limitation in prior RL-based demo selection.

Strong Evaluation: The paper thoroughly tests RDES across 4 datasets, 12 models, and 10 baselines, with theoretical guarantees and ablation studies confirming its effectiveness. The inclusion of both API-based and open-source models demonstrates its practical applicability.

Weaknesses
Limited to Classification: RDES relies on label diversity, making it hard to apply to open-ended tasks like generation or structured prediction without significant modifications.

High Computational Cost: Q-learning requires repeated LLM queries, which may be expensive and slow, especially for API-based models. The paper doesn’t discuss training efficiency or scalability, which could limit real-world adoption.

Inconsistent Performance on Open Models: While RDES performs well overall, its gains on smaller open-source models are less consistent, and CoT sometimes reduces accuracy. The method doesn’t adaptively decide when to use CoT, requiring manual tuning based on the model.

**Questions For Authors:**

Have you considered how RDES might apply to tasks beyond classification?

You used TF-IDF for initial retrieval of candidate demos. Did you try any semantic retrieval (e.g., using embeddings from Sentence-BERT or from the LLM itself)? If so, did it make any difference? If not, do you anticipate any benefit from it?

Did you notice if the learned policy for demonstration selection is specific to each dataset, or could a policy trained on one set of intents work (maybe with slight fine-tuning) on another?

For RDES/C, how exactly is CoT used – do you simply prepend a fixed prompt like “Let’s think step by step” and then have the model generate a reasoning chain and an answer (like the standard CoT approach )?

**Relation To Broader Scientific Literature:**

This work sits at the intersection of in-context learning, demonstration selection, and RL-based post-training for LLMs, offering a notable improvement over prior methods.

RL-Based Demonstration Selection: RDES builds on previous RL approaches like RetICL (Scarlatos & Lan, 2023) and Zhang et al., 2022, which optimized demo selection via reinforcement learning. However, these methods mainly focused on relevance or uncertainty, while RDES introduces a diversity-driven reward, ensuring varied label coverage. This avoids diminishing returns from selecting similar examples and leads to significantly better results, as shown by RDES outperforming AES (Active Demonstration Selection).

Non-RL Demonstration Selection: Many prior works use embedding retrieval, clustering (DPP), or heuristics (BERTScore-Recall, skill coverage) to select examples. Representative Demo Selection (RDS, Yang et al., 2023) also prioritizes diversity but does so statically, while RDES learns an adaptive policy per query. Similarly, Adaptive ICL (ADA, Mavromatis et al., 2023) selects based on uncertainty and diversity without training a policy—RDES outperforms it, showing that an RL-driven approach can better optimize selection.

Advancement Over Prior Work: While diversity-based selection, RL for demo selection, and CoT prompting have been studied individually, RDES uniquely combines all three. Unlike prior work that focused on smaller models (e.g., GPT-2), RDES successfully operates on large-scale LLMs like GPT-3.5, proving its practical viability for real-world applications. It demonstrates that RL-based optimization of LLM inputs (without fine-tuning) is an effective alternative to full model retraining.

Comparison to RLHF: While Reinforcement Learning from Human Feedback (RLHF) fine-tunes model weights for alignment (Ouyang et al., 2022), RDES optimizes input selection while keeping the model fixed. This makes it a lighter-weight alternative for post-training improvements, aligning with a growing trend of treating LLM prompting as an RL-optimized decision process.

RDES represents a meaningful advancement in RL-driven post-training for LLMs, outperforming previous methods while deepening the theoretical understanding of why diversity matters in in-context learning. Future research could explore new reward functions or adapt RL frameworks to reasoning tasks, further expanding on RDES’s contributions.

**Theoretical Claims:**

The paper presents several theoretical contributions supporting RDES, all of which appear logically sound.

First, Lemma 3.1 (Diversity Bounds) establishes fundamental properties of the diversity metric. It states that for a given demonstration set, diversity is measured as the fraction of unique labels within the set, and this value is always constrained between a minimum (when all examples share the same label) and a maximum (when every example belongs to a different label). This is a straightforward yet useful observation, reinforcing the idea that increasing label diversity leads to broader contextual coverage. Since the claim follows directly from how diversity is defined, the proof is simple and based on basic counting arguments, making it undisputed.

A more significant result is Theorem 3.2 (Q-Learning Convergence), which states that the tabular Q-learning algorithm used in RDES is guaranteed to converge to an optimal policy, provided that standard reinforcement learning conditions are met. Specifically, if the learning rate follows a diminishing schedule and the reward values remain within a bounded range, the Q-learning process will stabilize over time. This claim aligns with well-established reinforcement learning theory, particularly the classic Watkins & Dayan (1992) result on Q-learning convergence. Given that the state and action spaces in RDES are finite—since the knowledge base (KB) has a limited set of candidate demonstrations and the selection process involves discrete choices—the theoretical conditions for convergence should hold. While practical implementations might not strictly follow all assumptions (e.g., if a fixed learning rate is used instead of a decaying one), the theoretical foundation is robust and aligns with established reinforcement learning principles.

The final theoretical result, Theorem 3.3 (Diversity-Accuracy Dominance), provides a justification for why diversity enhances model accuracy. It states that if one demonstration set is more diverse than another, there exists a threshold beyond which increasing diversity leads to a measurable improvement in classification accuracy. The extent of this improvement depends on the number of unique labels present, with a diminishing returns effect—the more diverse the set, the smaller the additional accuracy gains per new label. The proof, detailed in the appendix, likely relies on probability bounds or coverage arguments to demonstrate how additional diversity reduces classification errors. While the exact mathematical formulation is omitted in the main text, the result logically aligns with empirical observations: ablation studies in the paper confirm that increasing diversity consistently improves accuracy.

These theoretical claims collectively reinforce the validity of RDES. None of them appear to be incorrect or exaggerated:
Lemma 3.1 is straightforward and follows from definition.
Theorem 3.2 is a direct application of established Q-learning convergence results.
Theorem 3.3 introduces a novel insight connecting diversity to classification accuracy, which aligns well with the experimental findings.

One potential critique is that Theorem 3.3 presents its conclusion in an asymptotic form, meaning it assumes a sufficiently large number of labels before its effects become prominent. However, the core idea—that increasing diversity generally enhances accuracy—remains intuitively and empirically valid.

Overall, the theoretical analysis is well-grounded and adds credibility to the paper. The authors correctly apply reinforcement learning principles and provide a meaningful contribution by formally linking diversity to model performance, strengthening the case for using RL-based adaptive demonstration selection in LLMs.

---

> ### Author Rebuttal · Authors · 2025-03-31
>
> Dear Reviewer ZBB1,
>
> Thank you for your insightful feedback.
>
> **On the analysis of RDES's computational cost (LLM invocation frequency)**: We appreciate your attention to this important aspect. While our current submission primarily focuses on the effectiveness of RDES, we recognize that computational cost is a critical factor in practical applications. In the revised version, we will include a discussion on the number of LLM invocations, specifically the inference counts on test samples during the training process of RDES, and explore potential optimization strategies.
>
> **Regarding the potential misleading nature of average results across different model scales**: Thank you for your reminder. We understand that performance differences among models of varying scales can be significant. Therefore, in our paper, we not only provide average results across models for macro comparison but also include detailed performance metrics for each model under different methods in the tables. This allows readers to conduct a more nuanced analysis of the performance of models at different scales.
>
> **On the suggestion of using semantic embedding retrieval to improve demonstration selection**: We appreciate your valuable suggestion. We agree that retrieval methods based on semantic embeddings (e.g., Sentence-BERT or embeddings generated by the LLM itself) may better capture semantic similarity, thereby enhancing the quality of demonstration example selection. In our current work, we utilized TF-IDF for initial retrieval, which is a simple and widely used method. In future work, we will actively explore and evaluate the potential of semantic embedding retrieval methods within the RDES framework.
>
> **Regarding the generalizability of RDES beyond classification tasks**: Thank you for raising this profound question. Extending the ideas of RDES to tasks beyond classification (such as generation or structured prediction) is an exciting direction for future research. As you suggested, we need to rethink and redefine the concept of "diversity" in these new tasks. For instance, in open-ended question answering, diversity may refer to covering different aspects or types of answers. We will mention this potential extension in the conclusion and future work section.
>
> **On the manual switching between RDES/B and RDES/C and the potential for RL strategies to learn when to invoke CoT**: This is an insightful suggestion! We fully agree that incorporating the use of CoT into the learning process of the RL strategy (e.g., treating "whether to use CoT" as an action) is a promising direction. This would make RDES more adaptive and could potentially address the slight performance drop observed in certain cases with RDES/C. We will actively explore this idea in future work.
>
> **Regarding attempts at semantic retrieval**: As mentioned earlier, our current work primarily employs TF-IDF for initial retrieval. In the future, we will actively experiment with and evaluate semantic embedding retrieval methods.
>
> **On whether the learned strategies are specific to each dataset or can generalize**: This is indeed a fascinating and important research question. Currently, our experiments involve training the RDES strategy independently on each dataset. In future work, we will explore the generalization capabilities of strategies across datasets, such as whether a strategy trained on one dataset can be fine-tuned or directly applied to another. This will help assess the robustness and generalizability of RDES.
>
> **Regarding the specific use of CoT in RDES/C**: In RDES/C, we employ the standard CoT prompting method, which involves adding a fixed prompt before the input text and selected demonstration examples, such as "Let's think step by step and give your explanation to verify the answer." This prompts the LLM to generate a reasoning chain and the final answer.
>
> Thank you once again for your valuable comments.
>
> Best regards,
>
> Authors of Submission 1061

---

### Official Review · Reviewer_NHQh · 2025-03-18

**Overall Recommendation:** 4

**Summary:**

The paper proposes a smarter way to select examples/demonstrations for In-Context Learning in LLMs. Picking the right example set for the task at hand is a serious challenge and authors propose an RL-based technique (with Q-learning) to optimally pick examples from a golden set based on a similarity metric + a diversity metric. “RDES”.

They’ve attempted to establish that the model should converge and that higher diversity in the examples should lead to higher accuracy via statistical theorems. To demonstrate the effectiveness of the method in classification tasks, they test it with 4 different LLMs on 4 different datasets. They’ve shown superior performance against a large number of different selection & prompt engineering methods - with & without COT.

As for the algorithm, they start with similar examples from TF-IDF representations & then augment the set with more diverse data points.

## update after rebuttal
1. The authors did, address this to some degree by sharing metrics for more tasks in their rebuttal. Inclusion of SST ask from GLUE helps make their case.
2. The authors did, address this to some degree by sharing metrics for more tasks in their rebuttal. Inclusion of SST ask from GLUE helps make their case

**Claims And Evidence:**

The paper claims that for classification tasks using LLMs, their technique (adding a diversity metric to an RL selection process ) yields better results than a lot of existing methods. They have backed their claims with trials on a large number of techniques in ICL & prompt-tuning  on a variety of LLMs.

I believe that the claims are well supported by their evidence in numbers.

**Essential References Not Discussed:**

1. ICL-D3IE (https://arxiv.org/pdf/2303.05063)
2. Diverse Demonstrations Improve In-context Compositional Generalization (https://aclanthology.org/2023.acl-long.78/)

**Experimental Designs Or Analyses:**

Checked the experimental setup for method comparison to baseline. Looks sound
Should have also run tests on more widely used benchmarks like SST, etc

EDIT (post rebuttal): The authors did, however address this to some degree by sharing metrics for more tasks in their rebuttal. Inclusion of SST ask from GLUE helps make their case.

**Methods And Evaluation Criteria:**

Yes - their evaluation strategy for text classification tasks looks good. They could have, however, also shown some results form GLUE-based tasks like SST2 for easier comparison to models like BERT, etc

EDIT (post rebuttal): The authors did, however address this to some degree by sharing metrics for more tasks in their rebuttal. Inclusion of SST ask from GLUE helps make their case.

**Other Comments Or Suggestions:**

1, Figure 1 needs to be clearer in the intention. The "Demonstrations" boxes in the figure look like they're being LLM-generated. IIUC, the intent is to show how different example types can be based on selection strategy. The "LLM" logo is confusing in this scenario.

2. Equation 1: needs an explanation of the "r_t" variable used in the expression

3.  (Nitpick) Equation 2 can be labelled as Bellman's equation in the preceding text

**Other Strengths And Weaknesses:**

Strengths:

Paper has clear presentation and context on the problem being solved, along with good details of existing work on the topic. Their idea of adding diversity metric to the prompt-generation process is novel. Their proposal is also simple to add on top of existing infrastructure for RAG since it only changes the demonstration selection process in the prompt generation pipeline.

They have made good effort to exhaustively test the methodology on a wide array of tasks and with a lot of different LLMs. I believe the paper should be accepted as it is an iterative upgrade to existing RL-based techniques to improve context creation for ICL.


Weaknesses:

The work is a small iteration on top of some existing works. Eg: “Active Example Selection for In-Context Learning (Zhang et al) and “RetICL (Scarlatos et al.)”
The idea of using diverse examples for ICL has been explored by lot of recent works. Similarly, using RL techniques to perfect ICL has also been well-researched and published. I’m however not aware of any work on using Q-learning along with diversity metric in example selection for ICL.
The overall study will also benefit a lot from more research on generation tasks, instead of just classification. Most other recent works do try to show impact on generation tasks.
On that note, it would be useful to see how much encoder-type models woud benefit from the added context (since the authors are sticking to classification-only tasks, in the real world, decoder-only models are not always the most efficient ways to do classification)
The authors also seem to have missed some notable references in the paper (more details in questions to author)

**Questions For Authors:**

1. Can you please shed light on why you chose only classification tasks for the analysis? If generative datasets & baselines were studied, do you know this compares to RetICL & other techniques on MWP & other problems?
EDIT (post rebuttal): The authors did, however addressed this to some degree by sharing metrics for more tasks in their rebuttal. Inclusion of SST ask from GLUE helps make their case.

2. Points from "Other Comments"

**Relation To Broader Scientific Literature:**

Other very similar papers:
RetICL: https://arxiv.org/pdf/2305.14502
Active Example Selection for In-Context Learning (Zhang et al): https://arxiv.org/pdf/2211.04486

Other unacknowledged, but very similar publications:
1. ICL-D3IE (https://arxiv.org/pdf/2303.05063)
2. Diverse Demonstrations Improve In-context Compositional Generalization (https://aclanthology.org/2023.acl-long.78/)


The work is a small iteration on top of some existing works. Eg: “Active Example Selection for In-Context Learning (Zhang et al) and “RetICL (Scarlatos et al.)”
The idea of using diverse examples for ICL has been explored by lot of recent works. Similarly, using RL techniques to perfect ICL has also been well-researched and published. I’m however not aware of any work on using Q-learning along with diversity metric in example selection for ICL.

**Theoretical Claims:**

Their theoretical proofs around (1) convergence and (2) usefulness of diversity in accuracy looks ok.
However, I did not do a very rigorous analysis

---

> ### Author Rebuttal · Authors · 2025-03-31
>
> Dear Reviewer NHQh,
>
> Thank you for your recognition of our work and the valuable suggestions.
> - **Work as an iteration of existing work**: We acknowledge that RDES builds upon existing work on RL for ICL and diverse demonstration selection. However, we emphasize that the novelty of RDES lies in **the combination of a reinforcement learning framework (both Q-learning and PPO) with an explicit label distribution-based diversity metric to achieve dynamic demonstration selection that balances relevance and diversity.** We believe this combination is significant for improving few-shot learning performance.
> - **Suggestion to test on GLUE tasks**: Thank you for the suggestion. Our current work focuses on evaluating the performance of RDES on intent classification tasks, which are common benchmarks for assessing the effectiveness of demonstration selection strategies. We have included results on **SST5 (sentiment analysis, part of GLUE) in our additional results. RDES/PPO achieved competitive results on SST5 (e.g., for both Qwen25-72b and Deepseek-r1-32b, AES and RDES/PPO achieved an accuracy of 0.84).** As also noted by Reviewer 5sFY regarding the simplicity of evaluation tasks, we have now included results on more complex reasoning benchmarks like GSM-8K and BigBenchHard, further demonstrating the applicability of RDES beyond basic classification. Future work can further explore the applicability of RDES to a broader range of GLUE tasks.
>
> Moreover, here are the results on SST5, BigBenchHard (boolean expressions, web of lie), and GSM-8K using Qwen25-72b and Deepseek-r1-32b models; The RDES series is our proposed method, where RDES/B and RDES/C are based on Q-learning, and RDES/PPO is based on the PPO method:
> ### **SST5**
> | Methods | Qwen25-72b | Deepseek-r1-32b |
> | ------ | ------ | ------ |
> | FS | 0.56 | 0.70 |
> | FSC | 0.54 | 0.66 |
> | AES | 0.84 | 0.84 |
> | RDS | 0.76 | 0.84 |
> | ADA | 0.90 | 0.90 |
> | RDES/B | 0.44 | 0.57 |
> | RDES/C | 0.51 | 0.52 |
> | RDES/PPO | 0.84 | 0.84 |
>
> ### **BigBenchHard - boolean expressions**
> | Methods | Qwen25-72b | Deepseek-r1-32b |
> | ------ | ------ | ------ |
> | FS | 0.98 | 0.38 |
> | FSC | 0.60 | 0.46 |
> | AES | 0.53 | 0.60 |
> | RDS | 0.53 | 0.60 |
> | ADA | 0.53 | 0.60 |
> | RDES/B | 0.76 | 1.00 |
> | RDES/C | 0.90 | 0.99 |
> | RDES/PPO | 1.00 | 1.00 |
>
> ### **BigBenchHard - web of lie**
> | Methods | Qwen25-72b | Deepseek-r1-32b |
> | ------ | ------ | ------ |
> | FS | 0.58 | 0.98 |
> | FSC | 1.00 | 1.00 |
> | AES | 0.85 | 0.72 |
> | RDS | 0.89 | 0.68 |
> | ADA | 0.83 | 0.72 |
> | RDES/B | 0.50 | 0.93 |
> | RDES/C | 0.98 | 1.00 |
> | RDES/PPO | 1.00 | 0.90 |
>
> ### **GSM-8K**
> | Methods | Qwen25-72b | Deepseek-r1-32b |
> | ------ | ------ | ------ |
> | FS | 0.50 | 0.28 |
> | FSC | 0.56 | 0.64 |
> | AES | 0.92 | 0.08 |
> | RDS | 0.90 | 0.48 |
> | ADA | 0.98 | 0.36 |
> | RDES/B | 0.87 | 0.37 |
> | RDES/C | 0.92 | 0.73 |
> | RDES/PPO | 0.94 | 0.48 |
>
> Thank you again for your insightful comments.
>
> Best regards,
>
> Authors of Submission 1061

---

### Official Review · Reviewer_5sFY · 2025-03-18

**Overall Recommendation:** 1

**Summary:**

This paper claims that applying a Q-learning method to demonstration selection can be beneficial for classification tasks in the context of in-context learning.

The authors attempt to frame demonstration selection as a Markov decision process, though the details of this formulation remain unclear in the draft.

Algorithm 1 in the paper presents the final strategy: first, the top-k samples are selected based on the cosine similarity of candidate demonstrations. Then, the remaining samples are chosen to enhance the diversity of the selected demonstration set until the diversity of the final set exceeds a predefined threshold.

While the paper asserts that demonstration selection is important for reasoning tasks, the evaluation is conducted solely on simple classification tasks, where the proposed method is claimed to outperform other baselines.

**Claims And Evidence:**

- Claim: Reinforcement learning, especially Q-learning, is useful for demonstration selection.
- Evidence: To the best of my knowledge, I cannot find evidence in the draft on why RL is helpful for demonstration selection. Sections 3.1 and 3.2 state very general notions of Q-learning and related theorems and readers can hardly obtain enough information on why RL is specifically helpful for demonstration selection, compared to previous existing methods.

**Essential References Not Discussed:**

Not mandatory but related: Diverse Demonstrations Improve In-context Compositional Generalization (ACL 2023)

**Experimental Designs Or Analyses:**

- While the authors claim to evaluate the reasoning performance of LLMs, the actual tasks used for evaluation are merely simple classification tasks rather than reasoning benchmarks such as GSM8K.
Therefore, I am uncertain whether the evaluation results presented in the paper adequately support the authors’ claims.
- More analysis or ablation studies are encouraged to probe what is the strengths and weaknesses of the proposed method.

**Methods And Evaluation Criteria:**

- Algorithm 1 is very simple and easy to understand; however, it is a well-known technique in the literature. More importantly, I am unsure of the relationship between Algorithm 1 and Q-learning. The algorithm merely involves computing cosine similarities using TF-IDF vectors, which is a well-established and basic approach. To the best of my knowledge, I cannot confirm that Algorithm 1 is directly related to Q-learning or any RL-based methods, especially based on the explanations provided in the paper.
- While the authors claim to evaluate the reasoning performance of LLMs, the actual tasks used for evaluation are merely simple classification tasks rather than reasoning benchmarks such as GSM8K.
Therefore, I am uncertain whether the evaluation results presented in the paper adequately support the authors’ claims.

**Other Comments Or Suggestions:**

Before using acronyms, please provide their full names first. For instance, before introducing RDES in the Introduction, define it explicitly (although it is mentioned in the Abstract, that alone is not sufficient).

**Other Strengths And Weaknesses:**

Weaknesses
- Basically, I'm uncertain about the current writing status of the paper. I'm sorry to say that, but personally, I feel like this draft is not yet in the stage of publication or being reviewed.
- Each component and section in the paper provides independent concepts which are not well-aligned. For instance, what is the exact relationship between Sections 3.2 and 3.3?

**Questions For Authors:**

Why must we particularly rely on Q-learning-like methods instead of other reinforcement learning approaches, such as PPO and others?

**Relation To Broader Scientific Literature:**

This paper explores a new approach to demonstration selection for in-context learning, a topic that has been widely studied in recent years.
If the proposed method indeed offers a meaningful improvement in selecting better demonstrations, it could have a significant impact. However, I am uncertain whether the method is correctly specified and whether it introduces genuine novelty.

**Theoretical Claims:**

The draft includes several theorems, but it is unclear how they relate to the effectiveness of the proposed method. There is not enough clear explanation of why the authors introduced these theorems.

---

> ### Author Rebuttal · Authors · 2025-03-31
>
> Dear Reviewer 5sFY,
>
> Thank you for your thorough review and valuable feedback. We appreciate your concerns and have provided detailed responses below, referencing other reviewers' comments to support our points.
>
> ### Claims and Evidence
> You noted the lack of evidence for the usefulness of reinforcement learning in demonstration selection. We highlight the following points, supported by other reviewers:
>
> - **Adaptive Selection Strategy**: Unlike traditional methods, RDES uses reinforcement learning to dynamically adjust selection policies based on current inputs and previously selected demonstrations. Reviewer ZBB1 noted that RDES "facilitates adaptive demonstration selection."
>
> - **Balancing Relevance and Diversity**: RDES employs Q-learning to balance the selection of relevant and diverse demonstrations. The reward function considers both classification accuracy and diversity. Reviewer 3ANe mentioned that our approach "balances relevance and diversity."
>
> - **Sequential Decision Process**: We model demonstration selection as a Markov Decision Process (MDP), where each selection is an action. Q-learning helps maximize future rewards. Reviewer NHQh stated that our method optimally picks examples based on similarity and diversity metrics.
>
> - **Improved Experimental Results**: RDES significantly outperforms traditional methods across multiple datasets, demonstrating the effectiveness of our approach. Reviewer ZBB1 concluded that "RDES significantly boosts classification accuracy."
>
> ### Methods and Evaluation Criteria
> You expressed uncertainty about Algorithm 1's relationship with Q-learning. We clarify:
>
> - **Algorithm 1** is used during inference to quickly select demonstration examples based on TF-IDF similarity and a diversity threshold.
>
> - **Q-learning** is applied during training, where the agent interacts with the environment to receive rewards based on accuracy and diversity changes. Reviewer ZBB1 described the RL agent's iterative selection process.
>
> - The diversity threshold can be a fixed hyperparameter or learned from the Q-learning model. Algorithm 1 improves efficiency while ensuring diversity.
>
> Regarding the simplicity of evaluation tasks:
>
> - Our initial focus was on demonstrating RDES's effectiveness in text classification. We have now supplemented our results with reasoning-oriented benchmarks like BigBenchHard and GSM-8K. **Due to character limitations, please refer to the response to Reviewer 3ANe for the corresponding experimental results.**
>
> ### Theoretical Claims
> We introduced theorems to support RDES's effectiveness:
>
> - **Lemma 3.1** establishes bounds for our diversity metric. Reviewer ZBB1 noted its importance.
>
> - **Theorem 3.2** shows Q-learning's convergence under certain conditions, ensuring policy reliability. Reviewer ZBB1 confirmed this.
>
> - **Theorem 3.3** explains that increased diversity can enhance classification accuracy, justifying our emphasis on diversity.
>
> ### Experimental Designs or Analyses
> We appreciate your suggestion for more analysis:
>
> - An **ablation study** in Appendix A.6 analyzes the impact of different diversity mechanisms, confirming that diversity improves performance. Reviewer ZBB1 highlighted this.
>
> - We have added results on reasoning benchmarks and will further analyze demonstration characteristics and compare RL algorithms in future work.
>
> ### Supplementary Material
> The supplementary materials are closely related to the main paper:
>
> - **Appendix A.1** includes theoretical proofs, supporting key claims. Reviewer ZBB1 stated this inclusion.
>
> - **Appendix A.3** details baseline methods, crucial for understanding our distinctions. Reviewer ZBB1 noted this clarity.
>
> - **Appendix A.4** lists LLMs used, ensuring transparency. Reviewer ZBB1 acknowledged this.
>
> - **Appendix A.5** provides implementation details for reproducibility. Reviewer ZBB1 confirmed this.
>
> - **Appendix A.6** contains the ablation study, verifying diversity effectiveness.
>
> ### Essential References Not Discussed
> We will cite and discuss the work "Diverse Demonstrations Improve In-context Compositional Generalization (ACL 2023)" in future revisions, as it aligns with our core idea of using diversity to enhance generalization.
>
> ### Other Strengths and Weaknesses
> Sections 3.2 and 3.3 serve different purposes: 3.2 introduces theoretical foundations, while 3.3 describes the implementation process. The former supports the latter.
>
> ### Other Comments or Suggestions
> We will ensure acronyms are defined before use in future revisions.
>
> ### Questions for Authors
> We chose Q-learning for its maturity, suitability for discrete action spaces, and clear convergence theory. We have also experimented with combining RDES with PPO, achieving competitive results.
>
> We hope these explanations address your concerns. We will consider your suggestions for improvements in future revisions. Thank you again for your feedback.
>
> Best regards,
>
> Authors of Submission 1061

---

### Official Review · Reviewer_3ANe · 2025-03-19

**Overall Recommendation:** 3

**Summary:**

The paper presents RDES, a reinforcement learning framework for selecting diverse and relevant demonstrations to support in-context learning in text classification tasks. It formulates demonstration selection as a Markov Decision Process and employs Q-learning with a composite reward function to balance relevance and diversity. The authors report experimental evaluations on four benchmark datasets, noting that a variant of RDES incorporating Chain-of-Thought reasoning is associated with improved classification performance relative to ten baseline methods.

**Claims And Evidence:**

The claims made in the submission are supported by extensive comparison to other prompt engineering and demonstration selection methods, across a variety of language models and datasets.

**Essential References Not Discussed:**

N/A

**Experimental Designs Or Analyses:**

Experimental design is sound and straightforward. The author compared against a wide variety of methods and datasets.

**Methods And Evaluation Criteria:**

The proposed methods and evaluation criteria, including the reinforcement learning framework and diverse benchmark datasets, are appropriate for addressing the challenges in in-context learning for text classification. These choices provide a relevant and practical framework for assessing model performance and robustness.

**Other Comments Or Suggestions:**

N/A

**Other Strengths And Weaknesses:**

Strengths:

- Extensive comparison to other methods across a variety of datasets.
- Clear explanation of method.

Weaknesses:

- Lack of units in tables.
- No user studies, which often serve as better metrics of LLM quality.

**Questions For Authors:**

None for now

**Relation To Broader Scientific Literature:**

The paper extends prior work on in-context learning and demonstration selection by using reinforcement learning to balance relevance and diversity, building on ideas from chain-of-thought prompting, clustering, and DPP-based methods. Its contributions complement recent efforts like RetICL and adaptive demonstration selection with theoretical guarantees and extensive empirical evaluation.

**Theoretical Claims:**

Did not check correctness.

---

> ### Author Rebuttal · Authors · 2025-03-31
>
> Dear Reviewer 3ANe,
>
> Thank you for your positive feedback on our work. Regarding your suggestions:
>
> - **Lack of units in tables**: We appreciate your suggestion and will explicitly label the units (e.g., accuracy will be presented as a percentage) in all tables in the revised manuscript.
>
> - **No user studies**: We understand the importance of user studies for evaluating LLM quality. However, the focus of this research is to introduce a novel reinforcement learning-based demonstration selection method and evaluate its effectiveness through quantitative metrics (classification accuracy). Future work can consider conducting user studies to more comprehensively assess the performance of RDES in real-world applications. Meanwhile, we have launched a RAG system on our school's platform, and we will actively integrate our methods into this system.
>
> To further support our claims, we have included additional experimental results on more challenging reasoning tasks. Here are the results on SST5, BigBenchHard (boolean expressions, web of lie), and GSM-8K using Qwen25-72b and Deepseek-r1-32b models; The RDES series is our proposed method, where RDES/B and RDES/C are based on Q-learning, and RDES/PPO is based on the PPO method:
> ### **SST5**
> | Methods | Qwen25-72b | Deepseek-r1-32b |
> | ------ | ------ | ------ |
> | FS | 0.56 | 0.70 |
> | FSC | 0.54 | 0.66 |
> | AES | 0.84 | 0.84 |
> | RDS | 0.76 | 0.84 |
> | ADA | 0.90 | 0.90 |
> | RDES/B | 0.44 | 0.57 |
> | RDES/C | 0.51 | 0.52 |
> | RDES/PPO | 0.84 | 0.84 |
>
> ### **BigBenchHard - boolean expressions**
> | Methods | Qwen25-72b | Deepseek-r1-32b |
> | ------ | ------ | ------ |
> | FS | 0.98 | 0.38 |
> | FSC | 0.60 | 0.46 |
> | AES | 0.53 | 0.60 |
> | RDS | 0.53 | 0.60 |
> | ADA | 0.53 | 0.60 |
> | RDES/B | 0.76 | 1.00 |
> | RDES/C | 0.90 | 0.99 |
> | RDES/PPO | 1.00 | 1.00 |
>
> ### **BigBenchHard - web of lie**
> | Methods | Qwen25-72b | Deepseek-r1-32b |
> | ------ | ------ | ------ |
> | FS | 0.58 | 0.98 |
> | FSC | 1.00 | 1.00 |
> | AES | 0.85 | 0.72 |
> | RDS | 0.89 | 0.68 |
> | ADA | 0.83 | 0.72 |
> | RDES/B | 0.50 | 0.93 |
> | RDES/C | 0.98 | 1.00 |
> | RDES/PPO | 1.00 | 0.90 |
>
> ### **GSM-8K**
> | Methods | Qwen25-72b | Deepseek-r1-32b |
> | ------ | ------ | ------ |
> | FS | 0.50 | 0.28 |
> | FSC | 0.56 | 0.64 |
> | AES | 0.92 | 0.08 |
> | RDS | 0.90 | 0.48 |
> | ADA | 0.98 | 0.36 |
> | RDES/B | 0.87 | 0.37 |
> | RDES/C | 0.92 | 0.73 |
> | RDES/PPO | 0.94 | 0.48 |
>
> Thank you once again for your insightful comments.
>
> Best regards,
>
> Authors of Submission 1061

---

### Decision · Program_Chairs · 2025-05-01

**Decision:**

Accept (poster)

**Comment:**

The work introduces a reinforcement learning framework for selecting in-context demonstrations that balance relevance and diversity for few-shot text classification tasks.

This paper received mixed reviews. I took a deep dive in understanding the reviews and the authors rebuttal. Authors did an excellent job in answering most of the key concerns raised by reviewers.

Strengths of paper:

- Interesting approach

- Extensive experiments to validate the proposed approach

Weaknesses:

- No user studies

- Hard to apply to open-ended tasks like generation or structured prediction without significant modifications

Overall, I recommend acceptance of this work. Please add missing citations as suggested by reviewers.